# LimiX-2M: Mitigating Low-Rank Collapse and Attention Bottlenecks in Tabular Foundation Models

**Yuanrui Wang** [* 1 2]  **Xingxuan Zhang** [* 2 3]  **Han Yu** [3]  **Mingchao Hao** [2]  **Gang Ren** [2]  **Hao Yuan** [2]  **Li Mao** [2]
**Yunjia Zhang** [2]  **Chun Yuan** [1]  **Peng Cui** [2 3]

## Abstract

Tabular foundation models (TFMs) increasingly rival tree ensembles, but their performance is often compute-inefficient: with standard affine scalar tokenization, each feature injects value variation through an essentially one-dimensional channel, and feature IDs/positional signals cannot increase within-feature value degrees of freedom, yielding weak early-layer value sensitivity and redundant hidden states. We present a unified *tokenize-and-route* framework for strong TFMs: **RaBEL** expands each scalar into compact localized RBF features (optionally exponent-gated) to improve conditioning and shallow-layer effective rank, while a reordered bidirectional block $S{\to}N{\to}F$ aligns computation with the readout by aggregating cross-sample context before feature mixing and using attention pooling. Together, these changes yield **LimiX-2M**, a 2M-parameter model that outperforms larger TabPFN-v2 and TabICL baselines on widely used tabular benchmarks while reducing training and inference costs. These results highlight value-aware tokenization and readout-aligned routing as key levers for improving the accuracy–efficiency trade-off in TFMs. Model checkpoints and inference code are available at https://github.com/limix-ldm-ai/LimiX.

## 1. Introduction

Recent advances in tabular foundation models—most notably TabPFN/TabPFN-v2 (Hollmann et al., 2025; 2022), TabICL (Qu et al., 2025), and LimiX (Zhang et al.,

2025), have narrowed, and in many regimes surpassed, long-standing baselines such as gradient-boosted decision trees (Chen & Guestrin, 2016; Ke et al., 2017; Prokhorenkova et al., 2018) and specialized deep tabular architectures (Somepalli et al., 2021; Arik & Pfister, 2021). These results are striking given the historical dominance of tree ensembles on medium-scale tabular tasks and the difficulty deep models have shown on irregular functions, heavy tails, and mixed data types (Grinsztajn et al., 2022).

Despite this progress, the input embedding layer remains a central limitation. The prevailing recipe maps each scalar cell through a single linear layer and augments it with a column identifier (e.g., positional or feature-ID embeddings), as in TabTransformer and FT-Transformer (Huang et al., 2020; Gorishniy et al., 2021). Systematic analysis indicates that such numeric embeddings are often overly restrictive and leave substantial performance on the table relative to more expressive encodings (Gorishniy et al., 2022). In our profiling, this design induces highly correlated activations early in the network: feature matrices in shallow layers can exhibit extremely low effective rank, sometimes collapsing to single-digit ranks on common benchmarks. This phenomenon implies significant parameter redundancy, suggesting that comparable performance could be achieved with a much smaller parameter budget. Moreover, it highlights untapped representational capacity: by rectifying this rank collapse to fully utilize the latent space, there is potential to unlock substantially richer feature representations.

We argue that the embedding layer for tabular FMs should play a greater role in introducing nonlinearity, enabling early representations to separate common tabular phenomena such as piecewise trends, local periodicity, quantization, heavy-tailed marginals, and heteroskedasticity. To this end, we propose RaBEL, a Radial Basis Embedding Layer that replaces the one-shot linear projection with a bank of localized nonlinear features. Classical theory and practice support RBF features as universal, localized approximators closely connected to kernel methods (Broomhead & Lowe, 1988; Park & Sandberg, 1991; Scholkopf & Smola, 2018; Rasmussen & Williams, 2006). Compared to direct linear embeddings, the localized nature of RBFs (i) yields diverse activation patterns across value regimes, (ii) improves con-

---

[1]Shenzhen International Graduate School, Tsinghua University [2]Stable AI [3]Department of Computer Science and Technology, Tsinghua University. Correspondence to: Peng Cui <cuip@tsinghua.edu.cn>, Chun Yuan <yuanc@sz.tsinghua.edu.cn>.

*Proceedings of the 43rd International Conference on Machine Learning*, Seoul, South Korea. PMLR 306, 2026. Copyright 2026 by the author(s).

ditioning of the first learned layer, and (iii) raises the effective rank of shallow representations without requiring many stacked layers to discover curvature ex post. The approach is complementary to periodic/Bochner-style mappings (e.g., random Fourier features) and can be extended or hybridized when periodicity is expected (Rahimi & Recht, 2007).

Beyond embeddings, we identify a second limitation in the permutation order of bidirectional attention used by existing tabular foundation models. In widely used designs (e.g., TabPFN-style or LimiX-style stacks), attention is typically arranged as feature-attention $\to$ sample-attention. This ordering introduces two problems. **(1)** In the very first layer, feature-level attention must integrate across columns based solely on raw values, before any column-level statistics or correlations have been established; this deprives attention of informative context and exacerbates low-rank collapse. **(2)** During prediction, many architectures consume only the target token from the final layer and thus the sample-level attention computed over features are effectively ignored, leading to weak training signals for parts of the network that do not directly influence the readout.

We address these issues by reordering the attention stack to sample-attention $\to$ (FFN) $\to$ feature-attention. The sample-attention phase at the input stage allows the model to aggregate column-level correlations and distributional statistics (e.g., moments, prevalence, missingness patterns) before engaging feature-level attention. An intermediate feed-forward network (FFN) then compresses and conditions these signals, after which the feature-attention phase learns inter-feature relations using richer, better-conditioned inputs. This permutation ensures that all attention computations contribute to the final prediction: information assembled at the sample-level directly shapes the feature-level representations that flow to the readout. This refined mechanism improves the capture of feature relationships and encourages the model to discover critical features, a capability we further analyze in the Section 5.3.

With the combination of RaBEL and the reordered sample-attention $\to$ FFN $\to$ feature-attention architecture, we introduce a 2M-parameter model, named **LimiX-2M**, that surpasses the 7M-parameter TabPFN-v2 baseline on mainstream benchmarks while cutting computational costs for both training and inference. These results indicate that principled nonlinear embeddings coupled with attention-order redesign can unlock better accuracy–efficiency trade-offs and more reliable scaling for tabular foundation models.

We summarize our contributions as follows.

1. We diagnose and quantify the low-rank collapse induced by linear+ID embeddings, and propose RaBEL, a compact RBF-based cell encoder that raises shallow-layer rank and improves conditioning.

2. We reveal a permutation-order pathology in standard bidirectional attention (feature-attention $\to$ sample-attention), and introduce a sample-attention $\to$ FFN $\to$ feature-attention stack that (i) establishes column-level statistics before feature aggregation and (ii) routes all attention signals to the readout.

3. We introduce LimiX-2M, a 2M-parameter model building on these two components that outperforms TabPFN-v2 with 7M-parameter and TabICL with 27M-parameter on most benchmarks while reducing both training and inference cost.

**Conflict of Interest Disclosure.** Some authors are affiliated with Stable AI. Stable AI leads the development of the LimiX model family, including LimiX-16M, one of the models evaluated in this paper.

## 2. Related Work

**Deep models for tabular data.** Gradient-boosted decision trees (GBDTs) such as XGBoost, LightGBM, and CatBoost have long dominated tabular prediction (Chen & Guestrin, 2016; Ke et al., 2017; Prokhorenkova et al., 2018). In response, specialized neural architectures have been proposed. TabNet performs attentive feature selection with interpretability (Arik & Pfister, 2021), TabTransformer applies self-attention over features—especially effective for categorical inputs (Huang et al., 2020), and SAINT augments feature-wise attention with intersample (row-wise) attention and contrastive pre-training (Somepalli et al., 2021). Despite these advances, broad evaluations still often find trees competitive on medium-scale benchmarks (Grinsztajn et al., 2022), underscoring the challenges of mixed data types and irregular target functions and motivating foundation-style approaches.

**Tabular foundation models.** TabPFN reframes tabular learning as in-context inference: a Transformer pre-trained on synthetic tasks consumes a small training set at inference and predicts without gradient updates (Hollmann et al., 2022; 2025). TabICL adopts a two-stage design that first builds per-sample representations with feature-then-row attention, followed by efficient in-context reasoning (Qu et al., 2025). LimiX treats a table as a joint distribution over features and missingness and uses masked modeling to support many tasks in one model (Zhang et al., 2025). Contemporary systems such as Mitra (Zhang et al., 2026) explore hybrid row–column attention with synthetic priors to improve cross-dataset generalization.

**Embedding strategies for numerical features.** A key design choice is how to encode continuous-valued cells. A prevalent recipe applies a single linear projection per numeric column, sometimes with column-identity embeddings,

as in FT-Transformer and TabTransformer (Gorishniy et al., 2021; Huang et al., 2020). Systematic analyses show that such encodings can be restrictive relative to more expressive schemes (Gorishniy et al., 2022). Effective remedies include (i) piecewise-linear encodings that partition value ranges into learnable segments and (ii) periodic encodings via sinusoidal features, conceptually related to random Fourier features (Gorishniy et al., 2022; Rahimi & Recht, 2007). Complementary self-supervised pre-training (e.g., VIME, MET) uses masked reconstruction to capture inter-feature dependencies prior to supervised fine-tuning (Yoon et al., 2020; Majmundar et al., 2022). Kernel-inspired alternatives based on radial basis functions (RBFs) offer localized receptive fields and universal approximation (Broomhead & Lowe, 1988; Park & Sandberg, 1991), closely connected to Gaussian RBF kernels and Gaussian processes (Scholkopf & Smola, 2018; Rasmussen & Williams, 2006). Learned RBF featurization thus provides localized nonlinear transformations that complement global periodic mappings like random Fourier features and can be naturally integrated into transformer-based tabular backbones.

## 3. The Low Rank Problem in Current Embeddings

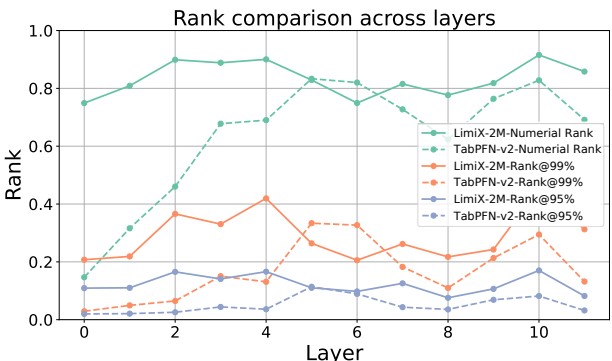

*Figure 1.* Rank comparison across layers of LimiX-2M and TabPFN-v2. The metric Rank@99% and Rank@95% represents the minimum number of SVD components required to take up 99% or 95% energy measured by singular values.

The current embedding strategy adopted by TabPFN-v2 is simply mapping each cell, i.e. each scalar, to the high-dimensional hidden space via a $1 \times p$ linear projection. Such a straightforward strategy implicitly leads to the low-rank problem: The hidden states output by transformer layers tend to be low-rank, especially early in the network. This could severely decrease the expressivity of the network, leading to potential performance degradation. In this section, we conduct both theoretical and experimental analyses to reveal the low-rank problem in TabPFN-v2.

First we theoretically prove the existence of the low-rank problem.

**Proposition 3.1.** *Assume the linear embedding strategy without positional embeddings, given a scalar sequence $x_1, x_2, ..., x_n \in \mathbb{R}$, the rank of the embedded input matrix is at most 2. For standard self-attention, the rank of the output matrix is at most 2. For multi-head attention with the number of heads $H$, the rank of the output is at most $H + 1$.*

The detailed proof can be referred to in Appendix A. Theorem 3.1 reveals that under the linear embedding strategy, the embedded input and the hidden states of shallower transformer layers could be extremely low-rank if no other non-linearity is induced. This is highly inefficient for such a high-dimensional hidden space.

To investigate the redundancy of latent representations, we analyze the spectral properties of hidden states across the OpenML-CC18 benchmark. Let $X \in \mathbb{R}^{B \times S \times D}$ denote the hidden states of a given layer, where $B$, $S$, and $D$ correspond to the batch size, sequence length, and hidden dimension, respectively. We reshape $X$ into a 2D matrix $\hat{X} \in \mathbb{R}^{(B \cdot S) \times D}$ to perform Singular Value Decomposition (SVD).

In Figure 1, we report the numerical rank, determined by the number of singular values exceeding a standard tolerance threshold. Additionally, we introduce Rank@99% and Rank@95%, defined as the minimum number of singular components required to preserve 99% and 95% of the cumulative spectral energy, respectively. All rank metrics are normalized by the hidden dimension $D$ for consistency.

To further validate the extent of representation redundancy, we conduct low-rank approximation experiments on TabPFN-v2. The architecture comprises a 12-layer network, where each layer contains three distinct modules: feature attention, sample attention, and a feed-forward network (FFN), totaling 36 modules. For each module during the forward pass, we apply truncated SVD to the input matrix, retaining only the top-$r$ singular values to reconstruct the low-rank approximation.

As shown in Table 1, the model exhibits negligible performance degradation compared to the full-rank baseline, even when the rank is truncated to $r = 50$ (approximately $25\%$ of the hidden dimension). Furthermore, the model maintains competitive AUC scores even at $r = 20$ ($\approx 10\%$ of the hidden dimension). These findings indicate a severe underutilization of the high-dimensional latent space, underscoring the necessity for more efficient embedding strategies.

In Appendix B, we theoretically show that under the standard affine scalar tokenizer, feature IDs/positional signals can separate columns but cannot increase the intrinsic degrees of freedom through which values enter the mode, yielding weak early-layer value sensitivity and redundant

representations. RaBEL breaks this value bottleneck by expanding each scalar into a compact localized basis before projection, which increases value diversity at the input and motivates mechanism-driven diagnostics (value-centered effective rank and Jacobian rank) that predict the observed accuracy–efficiency gains.

# 4. Method

## 4.1. Problem Setup and Notation

Let $X \in \mathbb{R}^{N \times D}$ denote a tabular dataset with $N$ samples (rows) and $D$ features (columns). We write $x_{i,j}$ for the scalar value at sample $i$ and feature $j$. Our goal is to map each cell $x_{i,j}$ to an embedding $e_{i,j} \in \mathbb{R}^d$ that is fed into a transformer with bidirectional attention operating along the sample and feature axes. We denote by $E \in \mathbb{R}^{N \times D \times d}$ the full tensor of cell embeddings, and by $L$ the number of attention blocks in the backbone.

This section introduces (i) a compact radial-basis embedding layer, **RaBEL**, that replaces the standard linear projection for scalar cells, and (ii) a reordered bidirectional attention block, from **Feature-Attention → Sample-Attention → FFN** (abbrev. F→S→N) to **Sample-Attention → FFN → Feature-Attention** (abbrev. S→N→F), that improves conditioning and ensures all attention computations contribute to the final prediction.

## 4.2. RaBEL: Radial-Basis Embedding Layer

Direct linear projection of numeric cells produces highly correlated early activations and very low effective rank in the first layers, which inhibits downstream capacity. RaBEL front-loads nonlinearity by expanding each scalar into a small bank of localized responses, followed by a light projection to the model dimension.

**Per-column normalization.** For numerical stability, we standardize each column using the z-score. Let $\bar{x}_j$ and $s_j$ denote the sample mean and standard deviation:

$$\tilde{x}_{i,j} = \frac{x_{i,j} - \bar{x}_j}{s_j + \epsilon}, \quad \bar{x}_j = \frac{1}{N}\sum_{i=1}^{N} x_{i,j}, \tag{1}$$

$$s_j = \sqrt{\frac{1}{N-1}\sum_{i=1}^{N}\left(x_{i,j} - \bar{x}_j\right)^2}. \tag{2}$$

Here, $\epsilon > 0$ is a small constant for numerical stability. All formulas below apply to either $x_{i,j}$ or $\tilde{x}_{i,j}$; we drop the tilde for brevity.

**RBF expansion.** For each feature $j$, we choose $M$ centers $\{c_{j,m}\}_{m=1}^{M}$ and bandwidths $\{\sigma_{j,m}\}_{m=1}^{M}$. Centers are initialized at empirical quantiles of $X_{\cdot,j}$; bandwidths are initialized proportional to local variability (e.g., interquartile range), and all parameters are subsequently learned end-to-end. Let the $m$-th component be $\kappa_{j,m} = \exp\big(-\frac{(x_{i,j}-c_{j,m})^2}{2\sigma_{j,m}^2}\big)$. The radial-basis feature map is

$$\phi_j(x_{i,j}) = \big[\kappa_{j,1}, \ldots, \kappa_{j,M}\big] \in \mathbb{R}^M. \tag{3}$$

**Projection to model dimension.** A shared linear projection with LayerNorm maps the expanded features to the model width:

$$e_{i,j} = \text{LN}\big(W_{\text{rbf}}\,\phi_j(x_{i,j}) + b_{\text{rbf}}\big) \in \mathbb{R}^d, \tag{4}$$

where $W_{\text{rbf}} \in \mathbb{R}^{d \times M}$ and $b_{\text{rbf}} \in \mathbb{R}^d$ are learned and shared across columns. For categorical columns, we use a standard entity embedding lookup into $\mathbb{R}^d$, followed by the same LayerNorm.

**Exponent-Gated RaBEL with *shared gates*.** Real-world tabular values span orders of magnitude and often exhibit heteroskedasticity. To make RaBEL *scale-aware* without losing locality, we introduce an **exponent gate** that conditions the *parameters* of the RBF bank via *shared* scalar gates applied uniformly across all $M$ basis functions.

*Exponent extraction (soft, differentiable).* Fix a base $\beta > 1$ (we use $\beta = 2$) and a small offset $\tau > 0$. Define the log-magnitude $\ell_{i,j} = \log_\beta(|x_{i,j}| + \tau)$. Let $\mathcal{B} = \{b_{\min}, \ldots, b_{\max}\} \subset \mathbb{Z}$ be a bounded set of exponent bins. We form a soft assignment via a temperature-controlled kernel:

$$\pi_{i,j}(b) = \frac{\exp\big(-(\ell_{i,j} - b)^2/T\big)}{\sum_{b' \in \mathcal{B}} \exp\big(-(\ell_{i,j} - b')^2/T\big)}, \quad b \in \mathcal{B}. \tag{5}$$

Each bin $b$ has a learnable embedding $u_b \in \mathbb{R}^h$, and we include a sign embedding $u_{\text{sgn}}$ for $\text{sgn}(x_{i,j})$. We obtain the *scale context* vector $z_{i,j}^{\exp} \in \mathbb{R}^{h+h_s}$ by concatenation:

$$z_{i,j}^{\exp} = \Big(\sum_{b \in \mathcal{B}} \pi_{i,j}(b)\,u_b\Big) \,\Big\|\, u_{\text{sgn}(x_{i,j})}. \tag{6}$$

*Shared gates on $(c, \sigma)$.* A small MLP produces two positive scalars per cell, shared across the RBF bank:

$$[\gamma_{i,j}^c, \gamma_{i,j}^\sigma] = \text{MLP}_{\text{shared}}(z_{i,j}^{\exp}) \in \mathbb{R}^2, \tag{7}$$

where positivity is enforced via Softplus. We compute the exponent-conditioned centers and widths:

$$c_{j,m}^{(\exp)} = \gamma_{i,j}^c\,c_{j,m}, \quad \sigma_{j,m}^{(\exp)} = \gamma_{i,j}^\sigma\,\sigma_{j,m}. \tag{8}$$

Let the gated RBF component be defined as:

$$\kappa_{j,m}^{(\exp)} = \exp\left(-\frac{(x_{i,j} - c_{j,m}^{(\exp)})^2}{2(\sigma_{j,m}^{(\exp)})^2}\right). \tag{9}$$

*Table 1.* AUC of TabPFN-v2 after low-rank approximation. We can see that the model consistently shows a competitive performance as the rank decreases to 20.

| Rank | 192 | 100 | 75 | 50 | 40 | 30 | 20 | 10 | 5 |
|---|---|---|---|---|---|---|---|---|---|
| AUC | 0.9177 | 0.9179 | 0.9175 | 0.9143 | 0.9100 | 0.9052 | 0.8985 | 0.8636 | 0.7674 |

The final gated feature vector is then:

$$\phi_j^{(\exp)}(x_{i,j}) = \left[\kappa_{j,1}^{(\exp)}, \ldots, \kappa_{j,M}^{(\exp)}\right]. \qquad (10)$$

Finally, we apply the projection $e_{i,j} = \mathrm{LN}(W_{\mathrm{rbf}}\phi_j^{(\exp)}(x_{i,j}) + b_{\mathrm{rbf}})$.

Exponent gating brings three benefits: (i) **Scale equivariance**: multiplying inputs by $\beta$ shifts $\ell$ by 1, thus the gate adapts the RBF bank across orders of magnitude, yielding unit- and scale-robust embeddings; (ii) **Heteroskedasticity robustness**: widths and amplitudes expand/contract in high/low variance regimes, maintaining useful locality of the bumps; (iii) **Better conditioning**: separating magnitude (via the exponent pathway) from pattern within a decade (via RBF responses) produces higher effective rank and smoother gradients in early layers. The soft assignment $\pi_{i,j}$ makes the module fully differentiable and avoids brittle hard binning. The computational overhead is small: an extra $O(|\mathcal{B}|)$ kernel evaluation and a tiny two-layer MLP.

### 4.3. Reordered Bidirectional Attention

Tabular transformers bidirectional-attention typically alternate attention across features (per sample) and across samples (per feature). Common stacks has the order of **FSN**(feature-attention $\rightarrow$ sample-attention $\rightarrow$ feed-forward). This forces the initial feature-level attention to integrate across columns using raw, weakly conditioned values. Moreover, the final prediction consumes only the target token, leaving other feature embeddings and thus the last sample-level attention computations underutilized.

We modify the bidirectional block by reordering its modules—without introducing any additional components—to **SNF**(sample-attention $\rightarrow$ feed-forward $\rightarrow$ feature-attention). In this design, the sample-attention layer first aggregates column-level statistics and cross-sample regularities, a lightweight feed-forward network conditions these signals, and the feature-attention layer then models inter-feature relations on better-conditioned inputs. The final embedding for prediction is obtained via attention pooling over all feature tokens, so every attention computation contributes directly to the output. This mechanism better captures feature interactions and promotes the discovery of key attributes, as analyzed in Section 5.3.

## 5. Experiments

### 5.1. Embedding Comparison

#### 5.1.1. MLP-BASED METHODS

We benchmark RaBEL against four baselines: No-embedding, MLP, PLE, and Periodic across diverse datasets (Table 7). For the embedding methods, we adopt a unified formulation where each scalar input $x_{j,i}$ is first transformed by a function $\phi_\theta : \mathbb{R} \rightarrow \mathbb{R}^d$, then flattened and mapped to the final dimension $e$ via a linear layer. The methods differ solely in $\phi_\theta$: **MLP** employs a point-wise FFN; **PLE** and **Periodic** utilize piecewise-linear and sinusoidal encodings (Gorishniy et al., 2022), respectively. Note that we modify PLE and Periodic to use *shared* linear projections to map encoded values to $\mathbb{R}^d$, rather than feature-specific ones. This adaptation accommodates varying feature counts, facilitating their subsequent application in foundation models. The **No-embedding** baseline directly projects the raw input $\mathbf{x}$ to $\mathbb{R}^{s \times e}$. Results are detailed in Table 2.

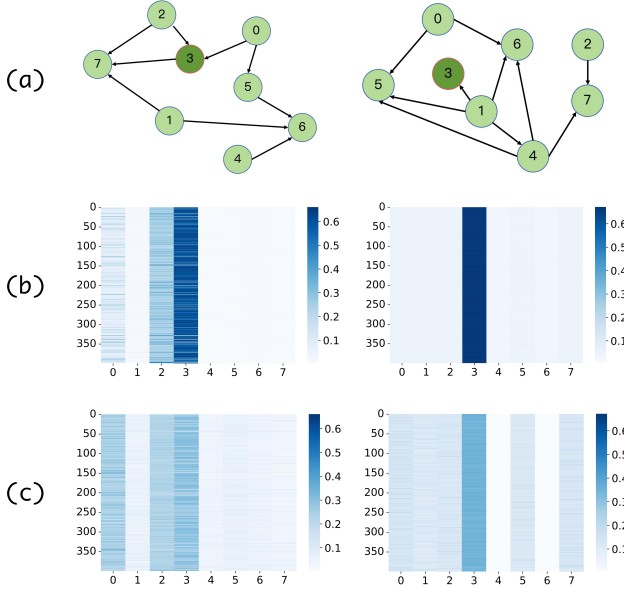

*Figure 2.* Visualization of attention scores (a) DAGs of the generated datasets. (b) Feature Attention Heatmap of FSN. (c) Feature Attention Heatmap of SNF. While FSN is dominated by self-attention, SNF demonstrates a broader attentional span that effectively targets neighboring features.

#### 5.1.2. TRANSFORMER-BASED METHODS

We integrate different embedding methods into a 2M-parameter transformer backbone with the same settings as

*Table 2.* Results for MLP with different embedding modules. The upper panel reports results for Metric1, and the lower panel reports results for Metric2. For each dataset, the arrow indicates whether higher or lower values are better for the corresponding metric.

| Metric1 | GC (↑) | CP (↑) | AC (↑) | CB (↑) | UK (↑) | BN (↑) | MA (↑) | CD (↑) | MH (↑) | CC (↑) | HS (↑) | SC (↑) | MV (↑) |
|---|---|---|---|---|---|---|---|---|---|---|---|---|---|
| MLP | 0.7537 | 0.8092 | 0.6560 | 0.8319 | 0.9850 | 0.7520 | **0.8896** | 0.4476 | 0.7811 | 0.6158 | 0.7497 | 0.1799 | **0.9999** |
| MLP-MLP | 0.8984 | 0.8496 | 0.8398 | 0.8924 | **0.9863** | 0.7518 | 0.8577 | 0.4768 | 0.8618 | 0.8947 | 0.8375 | 0.1749 | 0.9990 |
| MLP-PLE | 0.7393 | 0.8031 | 0.8262 | 0.8817 | 0.8416 | 0.7389 | 0.8609 | 0.2969 | 0.8281 | 0.8576 | 0.8279 | **0.1815** | 0.9730 |
| MLP-Periodic | 0.7817 | 0.8895 | 0.8054 | 0.8926 | 0.9821 | **0.7586** | 0.8625 | 0.4095 | 0.8306 | **0.9068** | 0.8286 | 0.1786 | 0.9994 |
| MLP-RaBEL | **0.9831** | **0.9582** | **0.9061** | **0.8979** | 0.9677 | 0.7580 | 0.8789 | **0.5124** | **0.8818** | 0.8998 | **0.8401** | 0.1813 | 0.9995 |

| Metric2 | GC (↑) | CP (↑) | AC (↑) | CB (↑) | UK (↑) | BN (↑) | MA (↑) | CD (↓) | MH (↓) | CC (↓) | HS (↓) | SC (↓) | MV (↓) |
|---|---|---|---|---|---|---|---|---|---|---|---|---|---|
| MLP | 0.3483 | 0.6756 | 0.6750 | 0.9677 | **0.9008** | 0.5757 | 0.9611 | 0.7218 | 0.4862 | 0.6072 | 0.5259 | 0.9051 | **0.0119** |
| MLP-MLP | 0.5393 | 0.7224 | 0.7870 | **0.9692** | 0.9008 | 0.5766 | 0.9657 | 0.7025 | 0.3863 | 0.3178 | 0.4238 | 0.9078 | 0.0321 |
| MLP-PLE | 0.2584 | 0.6890 | 0.7750 | 0.9677 | 0.5868 | 0.5596 | 0.9654 | 0.8143 | 0.4308 | 0.3697 | 0.4361 | **0.9042** | 0.1642 |
| MLP-Periodic | 0.3371 | 0.7759 | 0.7637 | 0.9677 | 0.8595 | **0.5788** | 0.9666 | 0.7463 | 0.4277 | **0.2991** | 0.4353 | 0.9058 | 0.0246 |
| MLP-RaBEL | **0.8090** | **0.8495** | **0.8377** | **0.9692** | 0.8678 | 0.5758 | **0.9668** | **0.6781** | **0.3573** | 0.3101 | **0.4203** | 0.9043 | 0.0221 |

*Table 3.* Results on BCCO-CLS with different embeddings.

| Embedding | AUC (↑) | Acc. (↑) | F1 (↑) |
|---|---|---|---|
| Transformer+MLP | 83.52 | 76.82 | 66.57 |
| Transformer+Periodic | 83.88 | 77.80 | 68.65 |
| Transformer+PLE | 84.66 | 77.68 | 67.74 |
| Transformer+RaBEL | **85.04** | **77.99** | **69.01** |

*Table 4.* Results on BCCO-REG with different embeddings.

| Embedding | $R^2$ (↑) | RMSE (↓) |
|---|---|---|
| Transformer+MLP | 0.7731 | 0.4043 |
| Transformer+Periodic | 0.6859 | 0.4321 |
| Transformer+PLE | 0.7410 | 0.4216 |
| Transformer+RaBEL | **0.7792** | **0.3964** |

*Table 5.* Rank comparison: LimiX-2M vs. 2M SNF baseline.

| Metric | 2M Baseline | LimiX-2M |
|---|---|---|
| Numerical Rank | 58.41 | **78.62** (+34.60%) |
| Rank@99% | 13.94 | **25.35** (+81.98%) |
| Rank@95% | 6.73 | **12.31** (+83.18%) |

in the previous section. Following the foundation-model training paradigm (training on generated data), we evaluate them on BCCO-CLS and BCCO-REG (Zhang et al., 2025). Results can be found in Table 3 and Table 4.

### 5.2. Comparison between LimiX-2M and 2M Baseline

To verify that the observed improvements stem specifically from RaBEL rather than the experimental optimization settings adopted from LimiX, we conduct a comparative analysis between LimiX-2M and a 2M-parameter baseline. While both models share identical training configurations and SNF layer sequences, they differ in their embedding mechanisms: the baseline utilizes a standard Linear projection, whereas LimiX-2M employs RaBEL. We report the rank comparison of the first three layers in Table 5, where the results demonstrate that LimiX-2M yields a significant rank boost in the shallow layers compared to the baseline. Furthermore, a comprehensive dataset-level performance comparison involving the baseline, LimiX-2M, and other competing models is detailed in Section C.7.

### 5.3. Toy Experiments of RBA

To empirically validate the capability of SNF in capturing latent feature dependencies, we conducted a toy experiment using a synthetic dataset generated from a Directed Acyclic Graph (DAG). Figure 2(a) illustrates the DAG structure with the target $y$ shaded. The corresponding feature attention maps are presented in Figure 2(b) and (c). These maps visualize the attention scores of the target $y$ with respect to other features, constructed by vertically stacking the attention vectors of all samples. In contrast to FSN, which relies heavily on self-attention, SNF exhibits reduced self-attention and effectively allocates attention to distinct features. Crucially, SNF assigns significantly higher attention scores to the direct causes of $y$ (e.g., Node 0), thereby confirming its superior performance in modeling feature interactions.

### 5.4. Comparison With SOTA Models

**Training Setting.** We construct our pre-training corpus using hierarchical Structural Causal Models (SCMs), following the data generation protocols established in the PFN series (Hollmann et al., 2022; 2025) and LimiX (Zhang et al., 2025). In each training episode, we first sample a random Directed Acyclic Graph (DAG) and functional mechanisms to define a specific joint distribution, from which synthetic data samples are subsequently drawn. The backbone of LimiX-2M is a 12-block Transformer architecture designed to capture both inter-sample and intra-feature dependencies. Each block is distinctively composed of three components in sequence: a **Sample-Attention** module, a **Feed-Forward Network (FFN)**, and a **Feature-Attention** module. The Sample-Attention mechanism facilitates interaction across different data samples (rows), while the Feature-Attention mechanism models the relationships between variables (columns) within a sample. The model is configured with a hidden embedding dimension of $d_{\text{model}} = 96$ and

*Table 6.* Aggregated average rank results. The benchmarks are categorized into **Classification** (BCCO-CLS, OpenML-cc18, PFN-CLS, TALENT-CLS, TabArena-CLS, TabZilla) and **Regression** (BCCO-REG, CTR23, PFN-REG, TALENT-REG, TabArena-REG). The reported values represent the average rank across AUC, Acc., and F1 for classification, and across $R^2$ and RMSE for regression. The best and second-best results are highlighted in red and blue, respectively. Missing entries ('-') indicate the model is incompatible with the specific task.

| Model | BCCO-CLS | BCCO-REG | CTR23 | OpenML-cc18 | PFN-CLS | PFN-REG | TALENT-CLS | TALENT-REG | TabArena-CLS | TabArena-REG | TabZilla |
|---|---|---|---|---|---|---|---|---|---|---|---|
| **Tree-based Method** | | | | | | | | | | | |
| CatBoost | 13.03 | 12.56 | 12.39 | 14.27 | 11.56 | 12.48 | 13.31 | 11.50 | 9.44 | 7.31 | 14.51 |
| LightGBM | 11.88 | 10.05 | 10.89 | 11.16 | 9.87 | 11.07 | 11.34 | 9.27 | 9.06 | 8.00 | 11.53 |
| XGBoost | 11.62 | 9.76 | 9.85 | 11.51 | 8.94 | 10.48 | 11.01 | 8.41 | 9.26 | 7.46 | 11.43 |
| **Deep-Learning Method** | | | | | | | | | | | |
| AutoInt | 21.68 | 19.14 | 18.95 | 19.87 | 23.53 | 20.13 | 22.88 | 20.18 | 20.31 | 19.15 | 18.38 |
| DANets | 21.80 | 30.49 | 30.00 | 19.89 | 21.24 | 28.83 | 20.94 | 29.44 | 22.43 | 30.46 | 22.48 |
| DCN-v2 | 18.55 | 15.84 | 15.82 | 19.94 | 24.74 | 16.35 | 19.86 | 16.70 | 20.28 | 16.73 | 18.93 |
| DNNR | - | 22.87 | 23.91 | - | - | 24.30 | - | 24.55 | - | 25.61 | - |
| ET | 15.96 | 12.15 | 11.97 | 17.07 | 14.66 | 12.59 | 17.08 | 10.96 | 13.20 | 11.35 | 17.02 |
| ExcelFormer | 16.71 | 13.67 | 13.89 | 14.21 | 14.62 | 15.43 | 15.49 | 14.71 | 16.15 | 15.38 | 14.27 |
| FT-Transformer | 15.63 | 15.51 | 15.95 | 17.74 | 19.66 | 15.69 | 17.75 | 16.93 | 20.86 | 23.08 | 17.49 |
| GrowNet | 29.11 | 28.87 | 28.65 | 27.94 | 28.90 | 27.54 | 28.48 | 28.14 | 26.85 | 28.39 | 27.00 |
| MLP | 20.23 | 19.05 | 17.12 | 16.01 | 17.83 | 16.00 | 18.30 | 18.79 | 19.21 | 19.61 | 18.47 |
| MLP-PLR | 17.40 | 17.80 | 16.92 | 17.23 | 21.29 | 16.11 | 19.67 | 17.20 | 19.82 | 18.54 | 17.01 |
| ModernNCA | 17.09 | 17.25 | 16.94 | 17.47 | 18.07 | 16.31 | 18.61 | 17.74 | 19.77 | 18.46 | 16.12 |
| NODE | 25.89 | 22.06 | 21.62 | 26.64 | 26.03 | 21.65 | 25.69 | 21.62 | 22.70 | 15.69 | 26.27 |
| RF | 13.45 | 13.45 | 12.80 | 14.89 | 12.26 | 14.93 | 14.85 | 11.60 | 10.43 | 11.00 | 13.33 |
| RealMLP | 13.33 | 5.92 | 6.32 | 9.36 | 12.33 | 7.46 | 10.17 | 7.12 | 20.16 | 8.00 | 10.33 |
| ResNet | 18.43 | 17.08 | 15.91 | 14.72 | 16.33 | 16.50 | 16.21 | 17.86 | 20.12 | 19.77 | 17.32 |
| SAINT | 19.32 | 18.55 | 17.89 | 25.98 | 23.38 | 17.72 | 21.54 | 19.45 | 24.45 | 20.25 | 20.75 |
| SNN | 22.31 | 27.22 | 26.77 | 22.49 | 24.56 | 26.20 | 24.05 | 26.52 | 22.78 | 26.69 | 21.62 |
| SwitchTab | 23.41 | 30.72 | 31.09 | 23.37 | 21.07 | 29.28 | 23.53 | 29.96 | 24.13 | 30.92 | 24.79 |
| T2G-Former | 15.80 | 14.91 | 15.05 | 16.49 | 18.53 | 14.98 | 17.50 | 15.35 | 18.85 | 20.54 | 16.05 |
| TANGOS | 18.25 | 17.74 | 18.14 | 14.91 | 17.21 | 17.00 | 17.60 | 18.37 | 18.13 | 16.54 | 16.16 |
| TabCaps | 24.51 | - | - | 22.34 | 21.39 | - | 21.53 | - | 20.67 | - | 23.53 |
| TabM | 12.21 | 5.40 | 6.53 | 10.75 | 10.56 | 7.61 | 10.57 | 6.44 | 14.80 | 6.69 | 10.15 |
| TabNet | 27.46 | 22.69 | 21.85 | 26.40 | 26.06 | 22.93 | 25.74 | 22.84 | 24.47 | 20.77 | 28.05 |
| TabR | 16.34 | 16.62 | 16.14 | 15.83 | 20.31 | 16.07 | 18.08 | 16.45 | 19.26 | 15.23 | 15.80 |
| TabTransformer | 22.17 | 30.65 | 30.03 | 24.05 | 23.52 | 30.35 | 23.46 | 29.60 | 21.08 | 30.33 | 22.51 |
| **Foundation Model** | | | | | | | | | | | |
| LimiX-16M (16.52M) | 2.71 | 4.28 | 4.61 | 4.99 | 2.87 | 5.65 | 4.02 | 3.58 | 4.78 | 3.46 | 5.63 |
| Mitra (75.67M) | 12.71 | 17.66 | 18.00 | 15.24 | 11.18 | 14.37 | 13.15 | 16.90 | 12.95 | 17.69 | 16.03 |
| TabICL (27.10M) | 9.42 | - | - | 10.04 | 8.79 | - | 7.77 | - | 8.27 | - | 10.57 |
| TabPFN-v2 (7.24M) | 10.58 | 6.39 | 8.44 | 8.56 | 7.03 | 8.50 | 7.61 | 7.02 | 7.11 | 5.92 | 9.57 |
| LimiX-2M (1.92M) | 7.31 | 6.52 | 7.65 | 5.83 | 5.71 | 5.89 | 6.53 | 7.02 | 5.10 | 4.54 | 6.58 |
| **Ensemble Method** | | | | | | | | | | | |
| AutoGluon | 9.39 | 5.13 | 5.89 | 7.79 | 7.47 | 7.59 | 7.98 | 5.78 | 7.02 | 3.46 | 8.62 |

employs $H = 6$ attention heads in each attention module, balancing computational efficiency with expressive power.

### 5.4.1. CLASSIFICATION

**Benchmarks.** We draw from six benchmark suites: TALENT-CLS (Ye et al., 2025), OpenML-CC18 (Bischl et al., 2017), PFN-CLS (Hollmann et al., 2025), TabZilla (McElfresh et al., 2023), TabArena-CLS (Erickson et al., 2025), and BCCO-CLS (Zhang et al., 2025). Following a common protocol, datasets with more than 50 000 training samples, over 10 000 features, or more than 10 target classes were excluded. After filtering, the final collection comprised 179 datasets from TALENT-CLS, 62 from OpenML-CC18, 29 from PFN-CLS, 27 from TabZilla, 33 from TabArena-CLS, and 106 from BCCO-CLS. For multiclass AUC and F1 calculations, we adopted a one-vs-one strategy.

**Metrics.** We evaluated performance using three metrics—AUC (Area Under the ROC Curve), Acc. (Accuracy), and F1 (F1-score).

### 5.4.2. REGRESSION

**Benchmarks.** We use five open-source benchmarks, TALENT-REG (Ye et al., 2025), PFN-REG (Hollmann et al., 2025), TabArena-REG (Erickson et al., 2025), CTR23 (Fischer et al., 2023), and BCCO-REG (Zhang et al., 2025). Applying the same filtering criteria as for the classification benchmarks yields 33 datasets from CTR23, 28 from PFN-REG, 13 from TabArena-REG, 99 from TALENT-REG, and 50 from BCCO-REG.

**Metrics.** We assessed regression performance using two metrics: $R^2$ (coefficient of determination) and RMSE (root mean squared error).

Please see Appendix C for more details.

### 5.4.3. RESULTS

We report the aggregated rank results in Table 6. To strike a favorable balance between inference efficiency and predictive quality, we trained a compact variant, LimiX-2M (1.92M). Despite its lightweight architecture, LimiX-2M demonstrates remarkable generalization capabilities, consistently achieving the second-best performance (highlighted in blue) across diverse classification and regression benchmarks, trailing only LimiX (16.52M). Notably, LimiX-2M significantly outperforms substantially larger foundation models, such as Mitra (75M) and TabICL (27M), as well as the specialized TabPFN-v2 (7M). Furthermore, it surpasses strong traditional ensembles like AutoGluon and XGBoost. These results compellingly validate that the proposed RaBEL framework and Reordered Bidirectional Attention mechanism enable high parameter efficiency without

compromising representation power. The detailed performance metrics for each individual dataset are provided in Section C.5.

## 5.5. Ablation Study

### 5.5.1. ABLATION ON MODULES AND RBA

We conduct ablation studies on two classification benchmarks TabZilla and TabArena to evaluate the Reordered Bidirectional Attention (RBA) and the SNF module design. First, results in Figure 3 validate the effectiveness of the RBA mechanism. Second, regarding the SNF architecture, we compare various sequences of attention and Feed-Forward Networks (FFN). Empirical results favor a sample-attention-first approach; specifically, interleaving the FFN between attention layers (SFN) optimally balances performance and parameter efficiency.

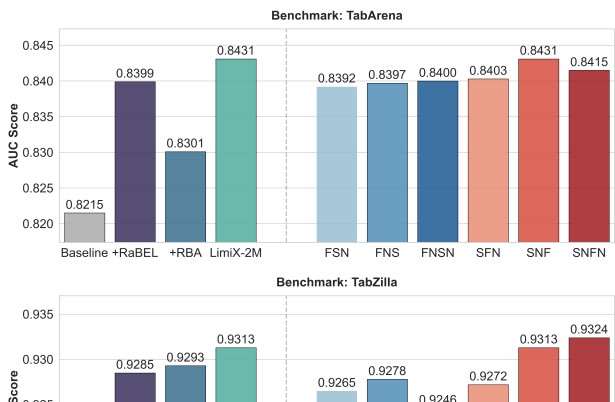
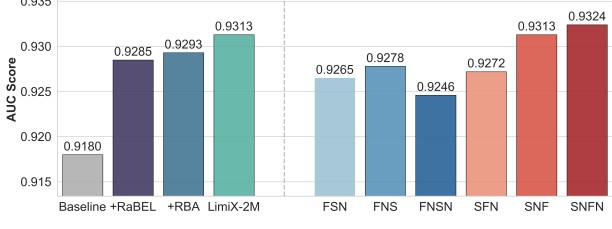

*Figure 3.* Comprehensive ablation studies on TabArena (top) and TabZilla (bottom) benchmarks. The visualization is divided into two distinct analyses: the Module Ablation (left) demonstrates the incremental performance gains (AUC) attributed to the integration of RaBEL and RBA modules into the baseline; the Structural Ablation (right) evaluates the impact of different topological orderings of components (Feature interaction, Structure, and Normalization) within the RBA block.

### 5.5.2. ABLATION ON RABEL

We perform an ablation study over the hyperparameters of RaBEL, sweeping settings such as the token embedding dimension, number of kernels, $\sigma$, initialization scheme, and kernel form. The results are reported in the Section C.3.

## 6. Conclusion

We presented **RaBEL**, a radial-basis embedding layer that replaces linear numeric encoders and resolves the early-

layer low-rank bottleneck by providing localized, scale-aware representations. We also revisited bidirectional attention and demonstrated that it improves conditioning and guarantees that every attention computation contributes to prediction. The resulting model, **LimiX-2M**, achieves superior accuracy with a substantially smaller parameter budget than prior tabular foundation models, while lowering compute. Future work will explore hybrid basis libraries, stronger self-supervised pretraining, and scaling LimiX-2M to broader domains and distribution shifts.

## Acknowledgments

This work is supported in part by the SSTIC Grant (KJZD20230923115106012, KJZD20230923114916032, GJHZ20240218113604008) and Tsinghua University-Siemens Joint Research Center (JCIIOT).

## Impact Statement

This paper contributes to the advancement of machine learning. While our work may have broader societal implications, we do not identify any specific consequences that warrant additional discussion beyond those commonly associated with progress in machine learning research.

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

# A. Proof of theoretical analyses

**Proof of Theorem 3.1.** The embedding modules consist of two $p$-dimensional parameter vector $\alpha, \beta \in \mathbb{R}^{p \times 1}$. Thus $x_i$ is embedded as $z_i^T = x_i \alpha^T + \beta^T$, and the embedded input matrix $z^T = [z_1, z_2, ..., z_n]^T \in \mathbb{R}^{n \times p}$. The rank of the matrix is obviously at most 2, since all its row vectors can be written as linear combinations of $\alpha$ and $\beta$.

The parameter matrix of self-attention are: $W_Q, W_K, W_V \in \mathbb{R}^{p \times p}$. Thus the attention score of query $x_i$ to $x_j$ is:

$$
\begin{aligned}
a_{ij} &= (x_i \alpha^T + \beta^T) W_Q ((x_j \alpha^T + \beta^T) W_K)^T \\
&= x_i x_j \alpha^T W_Q W_K^T \alpha + x_i \alpha^T W_Q W_K^T \beta + x_j \beta^T W_Q W_K^T \alpha + \beta^T W_Q W_K^T \beta \\
&= \lambda_1 x_i x_j + \lambda_2 x_i + \lambda_3 x_j + \lambda_4
\end{aligned}
\tag{11}
$$

Where $\lambda_1 = \alpha^T W_Q W_K^T \alpha,\ \lambda_2 = \alpha^T W_Q W_K^T \beta,\ \lambda_3 = \beta^T W_Q W_K^T \alpha,\ \lambda_4 = \beta^T W_Q W_K^T \beta$. Thus we have:

$$
\begin{aligned}
\hat{z}_i^T &= \sum_{j=1}^{n} \frac{\exp(a_{ij}/\sqrt{p})}{\sum_{j=1}^{n} \exp(a_{ij}/\sqrt{p})} \cdot (x_j \alpha^T + \beta^T) W_V \\
&= \left( \sum_{j=1}^{n} \frac{x_j \exp(a_{ij}/\sqrt{p})}{\sum_{j=1}^{n} \exp(a_{ij}/\sqrt{p})} \right) \alpha_V^T + \beta_V^T
\end{aligned}
\tag{12}
$$

Where $\alpha_V^T = \alpha^T W_V$ and $\beta_V^T = \beta^T W_V$. We can see that $z_i$ can also be written as the linear combination of two vectors $\alpha_V$ and $\beta_V$, thus the rank of the output is at most 2. For multi-head attention, the calculation of each head is exactly the same as standard self-attention. Use $W_O^{(h)}$ to denote the output projection matrix of head $h$. The output projection is:

$$
\begin{aligned}
\hat{z}_i &= \sum_{h=1}^{H} \hat{z}_{i,h}^T W_O^{(h)} = \sum_{h=1}^{H} (\mu_{i,h} \alpha_V^{(h)} + \beta_V^{(h)})^T W_O^{(h)} \\
&= \sum_{h=1}^{H} (\mu_{i,h} \alpha_O^{(h)} + \beta_O^{(h)})^T \\
&= \sum_{h=1}^{H} (\mu_{i,h} \alpha_O^{(h)})^T + \beta_O^T
\end{aligned}
\tag{13}
$$

Thus $\hat{z}_i$ can be written as the linear combinations of $H + 1$ vectors, indicating that the rank of the output is at most $H + 1$.

# B. Value-Sensitivity Collapse Under Standard Scalar Embeddings

This section explains a structural inefficiency in common TMFs: *even with feature-identity / positional signals, standard scalar tokenization injects each value $x_{i,j}$ through a severely low-dimensional channel*. The consequence is weak early-layer value sensitivity and high redundancy, which makes scaling width $d$ inefficient. We formalize this bottleneck and derive mechanism-driven diagnostics that our proposed RaBEL tokenizer and reordered bidirectional attention are designed to fix.

## B.1. Problem Setup and Notation

Let $X \in \mathbb{R}^{N \times D}$ denote a tabular dataset with $N$ samples (rows) and $D$ features (columns). We write $x_{i,j}$ for the scalar value at sample $i$ and feature $j$. Our goal is to map each cell $x_{i,j}$ to an embedding $e_{i,j} \in \mathbb{R}^d$ that is fed into a transformer with bidirectional attention operating along the sample and feature axes. We denote by $E \in \mathbb{R}^{N \times D \times d}$ the full tensor of cell embeddings, and by $L$ the number of attention blocks in the backbone.

**Feature-only signals (IDs/positions).** Many TFMs add a learned vector that depends only on the feature index $j$ (feature ID, positional signal, etc.). We introduce an auxiliary feature-only vector $a_j \in \mathbb{R}^d$ (set $a_j \equiv 0$ if absent). Crucially, $a_j$ carries *no information about the value $x_{i,j}$*.

## B.2. Baseline: Affine Scalar Tokenization Implies a One-Dimensional Value Channel

We analyze a widely used baseline for numerical cells: **direct linear projection** (often followed by LayerNorm), optionally plus feature-only signals. Concretely, define

$$e_{i,j}^{\lin} = \text{LN}\big(W_{\lin}\, x_{i,j} + b_{\lin}\big) + a_j, \qquad W_{\lin} \in \mathbb{R}^{d\times 1}, \; b_{\lin} \in \mathbb{R}^d. \tag{14}$$

Let $E^{\lin} \in \mathbb{R}^{N\times D\times d}$ denote the tensor whose $(i,j)$ entry is $e_{i,j}^{\lin}$.

The key point is not that IDs "do nothing"—they separate columns—but that they cannot increase the intrinsic dimension through which *values* enter the model. The right way to see this is to remove feature-only shifts and isolate within-feature value variation.

**Within-feature centering.** For any embedding tensor $E \in \mathbb{R}^{N\times D\times d}$, define the feature-$j$ slice $E_{:,j,:} \in \mathbb{R}^{N\times d}$ by stacking embeddings over samples. Define the feature-wise mean and centered embeddings:

$$\bar{e}_j = \frac{1}{N}\sum_{i=1}^{N} e_{i,j} \in \mathbb{R}^d, \qquad \tilde{e}_{i,j} = e_{i,j} - \bar{e}_j, \qquad \tilde{E}_j \in \mathbb{R}^{N\times d} : \; (\tilde{E}_j)_{i,:} = \tilde{e}_{i,j}^{\top}. \tag{15}$$

By construction, any feature-only vector $a_j$ cancels in $\tilde{e}_{i,j}$.

**Effective rank.** For a matrix $A$ with singular values $\{s_k\}$, define the effective rank

$$r_{\text{eff}}(A) = \exp\Big(-\sum_k p_k \log p_k\Big), \qquad p_k = \frac{s_k}{\sum_\ell s_\ell}. \tag{16}$$

We use $r_{\text{eff}}(\tilde{E}_j)$ to quantify the dimensionality of *value-induced* variation within feature $j$.

**Theorem B.1** (IDs do not increase within-feature value dimension). *Consider the baseline tokenizer*

$$e_{i,j}^{\lin} = \text{LN}\big(W_{\lin}\, x_{i,j} + b_{\lin}\big) + a_j, \qquad W_{\lin} \in \mathbb{R}^{d\times 1}, \; b_{\lin} \in \mathbb{R}^d, \; a_j \in \mathbb{R}^d, \tag{17}$$

*and define the pre-LayerNorm vector*

$$z_{i,j}^{\lin} = W_{\lin}\, x_{i,j} + b_{\lin} \in \mathbb{R}^d. \tag{18}$$

*For any feature $j$, let $\tilde{Z}_j^{\lin} \in \mathbb{R}^{N\times d}$ be obtained by centering $\{z_{i,j}^{\lin}\}_{i=1}^{N}$ across samples:*

$$(\tilde{Z}_j^{\lin})_{i,:} = \Big(z_{i,j}^{\lin} - \frac{1}{N}\sum_{i'=1}^{N} z_{i',j}^{\lin}\Big)^{\top}. \tag{19}$$

*Then* $\text{rank}(\tilde{Z}_j^{\lin}) \le 1$. *Equivalently, for any $i, i'$,*

$$z_{i,j}^{\lin} - z_{i',j}^{\lin} = W_{\lin}(x_{i,j} - x_{i',j}) \in \text{span}(W_{\lin}), \tag{20}$$

*and any additive feature-only vector $a_j$ cancels exactly under within-feature centering over $i$.*

*Proof.* Fix a feature index $j \in [D]$. We prove three claims: (i) the explicit centered form of $z_{i,j}^{\lin}$, (ii) the rank bound $\text{rank}(\tilde{Z}_j^{\lin}) \le 1$, and (iii) the cancellation of feature-only shifts $a_j$ under within-feature centering.

By definition (18),

$$\frac{1}{N}\sum_{i'=1}^{N} z_{i',j}^{\lin} = \frac{1}{N}\sum_{i'=1}^{N}\big(W_{\lin}x_{i',j} + b_{\lin}\big) = W_{\lin}\Big(\frac{1}{N}\sum_{i'=1}^{N} x_{i',j}\Big) + b_{\lin}.$$

Define the empirical mean of feature $j$ (over samples) as

$$\bar{x}_j = \frac{1}{N}\sum_{i=1}^{N} x_{i,j}. \tag{21}$$

Then the within-feature mean of $\{z_{i,j}^{\mathrm{lin}}\}$ is

$$\bar{z}_j \triangleq \frac{1}{N} \sum_{i=1}^N z_{i,j}^{\mathrm{lin}} = W_{\mathrm{lin}}\bar{x}_j + b_{\mathrm{lin}}. \tag{22}$$

Subtracting (22) from (18) yields, for each $i \in [N]$,

$$z_{i,j}^{\mathrm{lin}} - \bar{z}_j = \big(W_{\mathrm{lin}}x_{i,j} + b_{\mathrm{lin}}\big) - \big(W_{\mathrm{lin}}\bar{x}_j + b_{\mathrm{lin}}\big) = W_{\mathrm{lin}}(x_{i,j} - \bar{x}_j). \tag{23}$$

Thus every centered vector $z_{i,j}^{\mathrm{lin}} - \bar{z}_j$ lies in the one-dimensional subspace $\mathrm{span}(W_{\mathrm{lin}}) \subseteq \mathbb{R}^d$.

Let $w \triangleq W_{\mathrm{lin}} \in \mathbb{R}^{d \times 1}$, and define the centered scalar vector

$$v_j \triangleq \big[x_{1,j} - \bar{x}_j, \ x_{2,j} - \bar{x}_j, \ \ldots, \ x_{N,j} - \bar{x}_j\big]^\top \in \mathbb{R}^N. \tag{24}$$

Equation (23) implies that the $i$-th row of $\tilde{Z}_j^{\mathrm{lin}}$ equals

$$(\tilde{Z}_j^{\mathrm{lin}})_{i,:} = \big(z_{i,j}^{\mathrm{lin}} - \bar{z}_j\big)^\top = \big(w(x_{i,j} - \bar{x}_j)\big)^\top = (x_{i,j} - \bar{x}_j)\, w^\top.$$

Stacking over $i = 1, \ldots, N$ gives the exact matrix factorization

$$\tilde{Z}_j^{\mathrm{lin}} = v_j\, w^\top \in \mathbb{R}^{N \times d}. \tag{25}$$

An outer product of two vectors has rank at most 1 (and rank 0 if either vector is zero). Therefore,

$$\mathrm{rank}(\tilde{Z}_j^{\mathrm{lin}}) = \mathrm{rank}(v_j w^\top) \leq 1,$$

which proves the rank statement.

For any $i, i'$,

$$z_{i,j}^{\mathrm{lin}} - z_{i',j}^{\mathrm{lin}} = \big(wx_{i,j} + b_{\mathrm{lin}}\big) - \big(wx_{i',j} + b_{\mathrm{lin}}\big) = w(x_{i,j} - x_{i',j}) \in \mathrm{span}(w),$$

which is exactly (20).

Consider any embedding of the form $e_{i,j} = g_{i,j} + a_j$ where $a_j$ depends only on $j$ (not on $i$) and $g_{i,j} \in \mathbb{R}^d$ can be arbitrary (e.g., $g_{i,j} = \mathrm{LN}(z_{i,j}^{\mathrm{lin}})$ in (17)). Let

$$\bar{e}_j = \frac{1}{N} \sum_{i=1}^N e_{i,j} = \frac{1}{N} \sum_{i=1}^N (g_{i,j} + a_j) = \Big(\frac{1}{N} \sum_{i=1}^N g_{i,j}\Big) + a_j.$$

Then the centered embedding satisfies

$$e_{i,j} - \bar{e}_j = (g_{i,j} + a_j) - \Big(\frac{1}{N} \sum_{i'=1}^N g_{i',j} + a_j\Big) = g_{i,j} - \frac{1}{N} \sum_{i'=1}^N g_{i',j},$$

so $a_j$ cancels exactly. Applying this to (17) shows that any feature ID / positional vector $a_j$ does not contribute to within-feature variation over $i$. $\qquad\square$

**Interpretation (the "value bottleneck").** Theorem B.1 states that, before normalization, *each feature injects value variation through a one-dimensional channel*, independent of width $d$. IDs/positional signals can separate features (by shifting means across $j$), but they do not increase within-feature value degrees of freedom because they carry no dependence on $x_{i,j}$ and vanish under (15). This predicts low $r_{\mathrm{eff}}(\tilde{E}_j^{\mathrm{lin}})$ and highly correlated early activations.

**LayerNorm does not remove the bottleneck; it reshapes it locally.** LayerNorm is nonlinear, but locally around a batch it admits a first-order approximation. Let $J_{\mathrm{LN}}(z) \in \mathbb{R}^{d \times d}$ denote the Jacobian of LN at $z$. For small perturbations of $x_{i,j}$,

$$\delta\, \mathrm{LN}(z_{i,j}^{\mathrm{lin}}) \approx J_{\mathrm{LN}}(z_{i,j}^{\mathrm{lin}})\, W_{\mathrm{lin}}\, \delta x_{i,j}. \tag{26}$$

Thus, the *local* value-induced tangent directions remain confined to the span of $J_{\mathrm{LN}}(\cdot)W_{\mathrm{lin}}$, which is typically low-dimensional relative to $d$. In short: **increasing width can increase redundancy faster than it increases value sensitivity.**

## B.3. Consequences for Shallow Blocks: Low-Dimensional Value Sensitivity

Let $H^{(\ell)} \in \mathbb{R}^{N \times D \times d}$ denote the hidden tensor after $\ell \in \{0, 1, \dots, L\}$ attention blocks, with $H^{(0)} = E$. To measure how many independent directions feature $j$ can influence the representation, we define a value-sensitivity Jacobian.

**Value-sensitivity Jacobian.** For a fixed feature $j$, define

$$\mathcal{J}_j^{(\ell)} = \left[ \frac{\partial \mathrm{vec}(H^{(\ell)})}{\partial x_{1,j}}, \frac{\partial \mathrm{vec}(H^{(\ell)})}{\partial x_{2,j}}, \dots, \frac{\partial \mathrm{vec}(H^{(\ell)})}{\partial x_{N,j}} \right] \in \mathbb{R}^{(NDd) \times N}, \tag{27}$$

and use $r_{\mathrm{eff}}(\mathcal{J}_j^{(\ell)})$ as a diagnostic of *value sensitivity*.

**Proposition B.2** (Affine tokenization bottlenecks local value sensitivity (local linearization)). *Consider one numerical feature $j$, and let the scalar tokenizer be*

$$e_{i,j} = \mathrm{LN}(wx_{i,j} + b) + a_j,$$

*where $w, b, a_j \in \mathbb{R}^d$, and LayerNorm uses fixed gain and bias parameters.*

*Now consider a first-layer* sample-attention *block with $H$ attention heads applied to the sequence*

$$\{e_{1,j}, \dots, e_{N,j}\}.$$

*For any output position $t$, define the Jacobian of the output token with respect to the values of feature $j$ across the $N$ samples:*

$$K_{t,j} = \left[ \frac{\partial h_{t,j,:}^{(1)}}{\partial x_{1,j}}, \dots, \frac{\partial h_{t,j,:}^{(1)}}{\partial x_{N,j}} \right] \in \mathbb{R}^{d \times N}.$$

*Then there exists a linear subspace $U_j \subseteq \mathbb{R}^d$, independent of $t$ and of the specific perturbation direction, such that*

$$\dim(U_j) \leq 2H,$$

*and*

$$\frac{\partial h_{t,j,:}^{(1)}}{\partial x_{k,j}} \in U_j, \qquad \forall t, k \in \{1, \dots, N\}.$$

*Hence,*

$$\mathrm{rank}(K_{t,j}) \leq 2H \qquad \text{for every } t.$$

*If we stack the Jacobians for all output positions $t = 1, \dots, N$, we obtain the full first-layer value-sensitivity Jacobian*

$$J_j^{\mathrm{attn}} \in \mathbb{R}^{Nd \times N},$$

*which satisfies*

$$\mathrm{rank}(J_j^{\mathrm{attn}}) \leq 2HN.$$

*Moreover, if this attention sublayer is followed by any module that is linearized at the operating point, such as a first-order approximation of an FFN or a feature-mixing map L, then*

$$\mathrm{rank}(LJ_j^{\mathrm{attn}}) \leq \mathrm{rank}(J_j^{\mathrm{attn}}) \leq 2HN.$$

*In particular, this upper bound is* independent of the hidden width *$d$. Therefore, increasing width alone does not automatically increase the number of value-sensitive directions available in shallow layers.*

**Proof.** **Step 1: The tokenizer restricts value-dependent variation to at most a 2-dimensional affine subspace.**

Let

$$z(x) = wx + b.$$

Write

$$\bar{w} = \frac{1}{d}\mathbf{1}^\top w, \qquad \bar{b} = \frac{1}{d}\mathbf{1}^\top b,$$

and define the centered vectors

$$u = w - \bar{w}\mathbf{1}, \qquad v = b - \bar{b}\mathbf{1}.$$

Then the centered pre-normalized token is

$$z(x) - \frac{1}{d}(\mathbf{1}^\top z(x))\mathbf{1} = xu + v.$$

LayerNorm can therefore be written as

$$\mathrm{LN}(z(x)) = \beta + \frac{D_\gamma(xu + v)}{\sigma(x)},$$

where $D_\gamma = \mathrm{diag}(\gamma)$ and

$$\sigma(x) = \sqrt{\frac{1}{d}\|xu + v\|_2^2 + \varepsilon}.$$

Define

$$u' = D_\gamma u, \qquad v' = D_\gamma v.$$

Then

$$e_{i,j} = a_j + \beta + \alpha_i u' + \eta_i v',$$

for some scalars $\alpha_i, \eta_i$ depending on $x_{i,j}$.

Therefore, for a fixed feature $j$, all tokens $\{e_{i,j}\}_{i=1}^N$ lie in the affine subspace

$$c_j + \mathrm{span}\{u', v'\}, \qquad c_j := a_j + \beta.$$

Hence the *value-dependent part* of the tokenizer output is confined to a subspace of dimension at most 2.

This already shows that feature identity $a_j$ does not enlarge the value-sensitive subspace: it only shifts the affine offset.

**Step 2: Each attention head preserves this low-dimensional structure.**

Consider one attention head $h$. Let

$$p_h = W_V^{(h)} u', \qquad q_h = W_V^{(h)} v', \qquad c_h = W_V^{(h)} c_j.$$

Then the value vectors have the form

$$V_i^{(h)} = W_V^{(h)} e_{i,j} = c_h + \alpha_i p_h + \eta_i q_h.$$

Thus all value vectors for this head lie in the affine subspace

$$c_h + \mathrm{span}\{p_h, q_h\}.$$

The output of head $h$ at position $t$ is

$$o_t^{(h)} = \sum_{s=1}^N \omega_{ts}^{(h)} V_s^{(h)}, \qquad \sum_{s=1}^N \omega_{ts}^{(h)} = 1.$$

Substituting the expression for $V_s^{(h)}$ gives

$$o_t^{(h)} = c_h + \left(\sum_{s=1}^N \omega_{ts}^{(h)} \alpha_s\right) p_h + \left(\sum_{s=1}^N \omega_{ts}^{(h)} \eta_s\right) q_h.$$

Therefore, for every $t$, the output of head $h$ still lies in the same affine subspace

$$c_h + \text{span}\{p_h, q_h\}.$$

After concatenation and output projection, the full multi-head output can be written as

$$h_{t,j,:}^{(1)} = c_{\text{out}} + \sum_{h=1}^{H} A_t^{(h)} r_h + \sum_{h=1}^{H} B_t^{(h)} s_h,$$

where

$$r_h = W_O^{(h)} p_h, \qquad s_h = W_O^{(h)} q_h,$$

and $A_t^{(h)}, B_t^{(h)}$ are input-dependent scalars.

Hence all outputs belong to the affine subspace

$$c_{\text{out}} + U_j, \qquad U_j := \text{span}\{r_1, s_1, \ldots, r_H, s_H\},$$

with

$$\dim(U_j) \leq 2H.$$

**Step 3: The Jacobian columns must lie in this subspace.**

Since $h_{t,j,:}^{(1)}$ always belongs to the affine space $c_{\text{out}} + U_j$, any derivative of $h_{t,j,:}^{(1)}$ with respect to an input value $x_{k,j}$ must belong to the corresponding direction space $U_j$. Therefore,

$$\frac{\partial h_{t,j,:}^{(1)}}{\partial x_{k,j}} \in U_j, \qquad \forall t, k.$$

This immediately implies

$$\text{rank}(K_{t,j}) \leq \dim(U_j) \leq 2H.$$

Stacking over all output positions $t$ yields the bound

$$\text{rank}(J_j^{\text{attn}}) \leq 2HN.$$

Finally, composing with any linearized downstream map $L$ cannot increase rank:

$$\text{rank}(L J_j^{\text{attn}}) \leq \text{rank}(J_j^{\text{attn}}) \leq 2HN.$$

**Interpretation.**   The proposition formalizes the key point behind *low value sensitivity*:

- the standard affine scalar tokenizer injects the value of a numerical feature through a very small number of directions;

- self-attention can mix samples, but it cannot magically create a large number of new value-sensitive directions from such a narrow input channel;

- as a result, the shallow-layer Jacobian with respect to feature values remains low-rank even when the hidden width $d$ is very large.

Therefore, simply widening the model does not remove the bottleneck. To improve value sensitivity, one must enrich the *value-dependent input geometry itself*, rather than only increasing hidden dimension.

**Implication.**   Proposition B.2 connects Theorem B.1 to deep computation: although attention is nonlinear, the *tangent space* through which a single feature's values influence shallow representations remains low-dimensional under (14). This provides a mechanism-level explanation for two empirical phenomena commonly observed in TFMs: (i) early activations exhibit low effective rank (high redundancy), and (ii) widening $d$ yields diminishing returns unless the tokenizer increases the intrinsic value-sensitive subspace.

### B.4. How RaBEL Breaks the Bottleneck

RaBEL replaces the scalar identity map in (14) with a compact radial-basis expansion $\phi_j(x_{i,j}) \in \mathbb{R}^M$ and then projects to $\mathbb{R}^d$:

$$e_{i,j} \;=\; \mathrm{LN}\big(W_{\mathrm{rbf}}\,\phi_j(x_{i,j}) + b_{\mathrm{rbf}}\big) \in \mathbb{R}^d, \tag{28}$$

The pre-LN term $W_{\mathrm{rbf}}\phi_j(x_{i,j})$ can vary along multiple directions as $x_{i,j}$ changes, allowing within-feature value variation to have substantially higher effective rank before reaching the backbone. This is the intended mechanism: **inject localized nonlinearity at the tokenizer so early layers are value-aware without requiring depth to manufacture curvature.**

### B.5. Design Takeaway

IDs/positional signals help the model identify *which* feature a token came from, but they do not increase the degrees of freedom through which *values* enter the network. Under standard affine tokenization, within-feature value variation is inherently low-dimensional (Theorem B.1), leading to low-dimensional local value sensitivity in shallow layers (Proposition B.2) and substantial redundancy. RaBEL and the reordered bidirectional attention block introduced in Section 4 are designed to address complementary failure modes: RaBEL expands the value-sensitive subspace at the tokenizer, and the reordered block improves conditioning and routes cross-sample computation into the representation used for prediction.

## C. Additional Experimental Details and Results

### C.1. Baseline Setup

All baseline results follow the TALENT benchmark protocol. Each dataset is split into 64/16/20 train/val/test; hyperparameters are tuned on the validation split with Optuna (100 trials per method–dataset pair), with early stopping based on validation accuracy for classification and RMSE for regression. The selected configuration is then retrained and evaluated over 15 random seeds, and the final result is reported as the seed average. Importantly, TALENT uses method-specific search spaces, not a single shared one. For foundation models such as TabPFN-v2, we evaluate on the full dataset. We also strictly unify preprocessing and ensemble count across methods, while otherwise using each method's standard settings. Thus, the comparisons are controlled and reproducible.

### C.2. Dataset Statistics for MLP-based Experiments

We collected datasets for both classification and regression tasks and summarized their key statistics, including the number of training samples, total features, categorical features, number of classes (for classification datasets), and missing value rates. These results are presented in Table 7.

*Table 7.* Statistics of datasets. These datasets cover varying sample sizes, tasks, data distributions, and levels of missingness. AUC: area under the ROC curve; Acc.: classification accuracy; $R^2$: coefficient of determination; RMSE: root mean squared error.

| Dataset | GC | CP | AC | CB | UK | BN | MA | CD | MH | CC | HS | SC | MV |
|---|---|---|---|---|---|---|---|---|---|---|---|---|---|
| #train samples | 205 | 696 | 7000 | 5455 | 282 | 44236 | 30305 | 726 | 9643 | 671 | 15129 | 36421 | 28537 |
| #features | 9 | 3 | 7 | 95 | 5 | 9 | 4 | 12 | 15 | 8 | 20 | 4 | 10 |
| #cate features | 7 | 1 | 5 | 2 | 0 | 7 | 0 | 1 | 2 | 0 | 4 | 1 | 3 |
| Metric1 | AUC | AUC | AUC | AUC | AUC | AUC | AUC | $R^2$ | $R^2$ | $R^2$ | $R^2$ | $R^2$ | $R^2$ |
| Metric2 | Acc. | Acc. | Acc. | Acc. | Acc. | Acc. | Acc. | RMSE | RMSE | RMSE | RMSE | RMSE | RMSE |
| #classes | 7 | 4 | 2 | 2 | 5 | 3 | 2 | – | – | – | – | – | – |
| #missing ratio | 0.2366 | 0 | 0.2892 | 0 | 0 | 0 | 0 | 0.00056 | 0 | 0 | 0 | 0 | 0 |

**Dataset abbreviations.** GC: guitar-chord-finger-positioning; CP: companion-plants; AC: ad-click-prediction-dataset; CB: company_bankruptcy_prediction; UK: user-knowledge; BN: BNG(cmc); MA: malware-analysis-datasets-pe-section-headers; CD: 1000-Cameras-Dataset; MH: miami_housing; CC: concrete_compressive_strength; HS: house_sales; SC: seattlecrime6; MV: mv.

### C.3. Ablation on RaBEL

We conducted a comprehensive ablation study to investigate the impact of key hyperparameters and design choices of RaBEL on model performance (Tables 8–13). regarding model capacity, we observe that an embedding dimension of 32 achieves the optimal trade-off (Table 8), while increasing the number of kernels consistently improves representation power, peaking

at 64 kernels (Table 9). For kernel configuration, a fixed bandwidth $\sigma = 1.0$ with uniformly distributed kernels proves most effective, outperforming learnable $\sigma$ and random distributions (Table 10, 12, 13). Most notably, the initialization strategy plays a critical role; as shown in Table 11, orthogonal initialization yields a substantial performance gain (reaching 89.03% AUC) compared to standard Xavier (Glorot & Bengio, 2010) or Kaiming (He et al., 2015) initializations, highlighting the importance of orthogonality in the exponent embedding.

*Table 8.* Impact of token embedding dimension

| dim | AUC | Acc. | F1 |
|---|---|---|---|
| 16 | 84.51 | 77.68 | 68.32 |
| 32 | **85.17** | **77.95** | **69.12** |
| 64 | 84.78 | 77.82 | 68.71 |

*Table 9.* Impact of number of kernels

| n kernels | AUC | Acc. | F1 |
|---|---|---|---|
| 16 | 84.20 | 77.12 | 68.53 |
| 32 | 84.76 | 77.68 | 68.88 |
| 64 | **85.19** | **77.95** | **69.08** |

*Table 10.* Impact of $\sigma$ value

| $\sigma$ | AUC | Acc. | F1 |
|---|---|---|---|
| 0.5 | 84.66 | 77.62 | 68.85 |
| 1.0 | **84.75** | **77.93** | **69.01** |
| 2.0 | 84.67 | 77.44 | 68.04 |

*Table 11.* Exponent embedding init method

| method | AUC | Acc. | F1 |
|---|---|---|---|
| orthogonal | **89.03** | **77.67** | **68.95** |
| xavier | 84.80 | 77.57 | 68.53 |
| kaiming | 84.68 | 77.44 | 68.27 |

*Table 12.* Impact of kernels form

| form | AUC | Acc. | F1 |
|---|---|---|---|
| random | 84.43 | 77.67 | 68.55 |
| uniform | **84.99** | **77.93** | **69.07** |

*Table 13.* Learnable vs. Fixed $\sigma$ (init $\sigma = 1$)

| mode | AUC | Acc. | F1 |
|---|---|---|---|
| learn | 84.65 | 77.59 | 68.53 |
| fix | **84.81** | **77.93** | **69.02** |

### C.4. Targeted Comparison with TabPFN-v2.5

Our main experimental campaign was conducted in September 2025, and LimiX-2M was released in October 2025. At that time, TabPFN-v2.5 (Grinsztajn et al., 2026) was not yet publicly available. The TabPFN-v2.5 technical report and model release appeared later in November 2025. Therefore, our primary comparisons in the main paper focus on the publicly available tabular foundation model baselines at the time of our main experiments, including TabPFN-v2 and TabICL.

After TabPFN-v2.5 became available, and in response to reviewer interest in this recent baseline, we conducted an additional targeted comparison on the BCCO benchmark under the same evaluation protocol. This experiment is intended as a post-release sanity check rather than a full re-benchmarking across all datasets and seeds. As shown in Table 14, LimiX-2M remains competitive with TabPFN-v2.5: it achieves higher AUC and accuracy on BCCO-CLS, and obtains comparable regression performance on BCCO-REG. These results suggest that the main empirical observations of this paper are not solely due to comparison with an earlier PFN baseline.

### C.5. Results on Open-Source Benchmarks

In this section, we provide the complete, dataset-level performance metrics for all open-source benchmarks evaluated in this study. While the main text reports aggregated performance, the following tables present the specific results for both classification and regression tasks across the BCCO, OpenML-cc18, PFN, TabArena, TabZilla, TALENT, and CTR23 benchmarks. These detailed breakdowns allow for a granular analysis of model performance on individual datasets.

### C.6. Inference Speed Comparison

We evaluate inference efficiency using a synthetic classification dataset comprising 900 samples and 60 features. All results are reported as the average of three runs on an AMD EPYC 9354 CPU and an NVIDIA RTX 4090 GPU (Table 26). LimiX-2M achieves a remarkable inference time of 171.40 ms on GPU, outperforming TabPFN-v2 by $\approx 2\times$ and TabICL by $> 10\times$. Furthermore, LimiX-2M maintains a substantial lead in CPU environments against heavy models like Mitra, confirming its practical deployability.

*Table 14.* Performance comparison on BCCO-CLS and BCCO-REG.

| Model | BCCO-CLS | | BCCO-REG | |
| --- | --- | --- | --- | --- |
| | AUC | Acc. | $R^2$ | RMSE |
| TabPFN-v2.5 | 0.852 | 0.778 | 0.780 | 0.391 |
| (Real) TabPFN-v2.5 | 0.853 | 0.779 | 0.777 | 0.392 |
| MiniX | 0.858 | 0.787 | 0.785 | 0.392 |

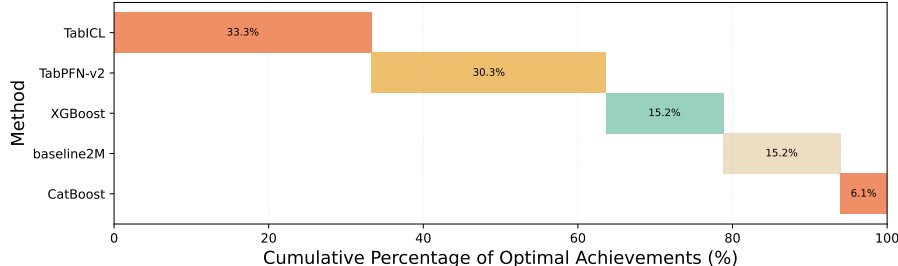

*Figure 4.* Cumulative Percentage of Optimal Achievements of Baseline2M on TabArena.

## C.7. Fine-grained Dataset-level Comparison

We conducted a fine-grained dataset-level comparison on the TabArena (classification) and CTR23 (regression) benchmarks to evaluate the number of datasets where each model achieves the leading performance. Our comparison set includes established baselines such as TabPFN-v2, TabICL, Mitra, XGBoost, and CatBoost.

Notably, we exclude LimiX from this specific visualization. Since LimiX achieves state-of-the-art results on the vast majority of datasets, including it would obscure the relative performance dynamics between LimiX-2M and the other baselines.

As shown in Figures 4 to 7, Baseline-2M rivals XGBoost but slightly trails TabICL and TabPFN-v2, whereas LimiX-2M consistently surpasses these baselines.

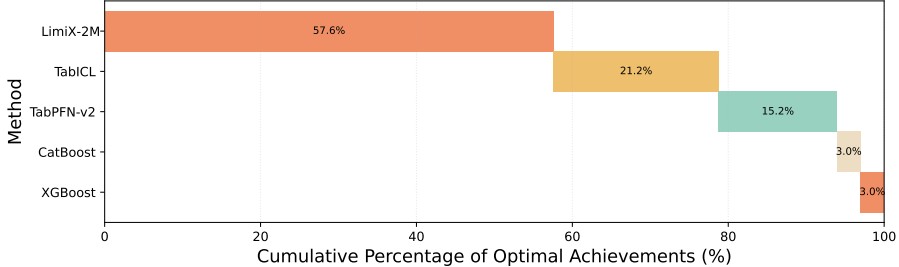

*Figure 5.* Cumulative Percentage of Optimal Achievements of LimiX-2M on TabArena.

*Table 15.* Classification results on **BCCO-CLS**, sorted by mean AUC in descending order, with the parameter counts of all foundation models highlighted in blue.

| | BCCO-CLS | | | | | |
| | Mean | | | Rank | | |
| Model | AUC (↑) | Acc. (↑) | F1 (↑) | AUC (↓) | Acc. (↓) | F1 (↓) |
|---|---|---|---|---|---|---|
| LimiX-16M (16.52M) | 0.871 | 0.804 | 0.731 | 2.679 | 2.387 | 3.075 |
| LimiX-2M (1.92M) | 0.858 | 0.787 | 0.701 | 6.406 | 6.689 | 8.830 |
| TabICL (27.10M) | 0.847 | 0.768 | 0.672 | 7.623 | 9.226 | 11.396 |
| AutoGluon | 0.846 | 0.771 | 0.677 | 8.792 | 8.943 | 10.425 |
| TabPFN-v2 (7.24M) | 0.843 | 0.772 | 0.679 | 9.575 | 10.274 | 11.896 |
| XGBoost | 0.834 | 0.762 | 0.674 | 11.160 | 11.491 | 12.217 |
| Mitra (75.67M) | 0.836 | 0.764 | 0.664 | 11.566 | 12.321 | 14.236 |
| LightGBM | 0.832 | 0.763 | 0.678 | 12.189 | 11.406 | 12.057 |
| TabM | 0.827 | 0.763 | 0.666 | 12.660 | 11.670 | 12.292 |
| RF | 0.829 | 0.756 | 0.652 | 12.802 | 13.104 | 14.453 |
| CatBoost | 0.829 | 0.757 | 0.664 | 13.358 | 12.472 | 13.264 |
| ET | 0.825 | 0.745 | 0.618 | 13.953 | 15.557 | 18.368 |
| RealMLP | 0.824 | 0.759 | 0.673 | 14.717 | 13.245 | 12.019 |
| FT-Transformer | 0.813 | 0.744 | 0.642 | 15.189 | 15.698 | 16.009 |
| T2G-Former | 0.808 | 0.742 | 0.646 | 16.255 | 16.009 | 15.151 |
| ModernNCA | 0.815 | 0.752 | 0.658 | 17.123 | 17.292 | 16.868 |
| MLP-PLR | 0.804 | 0.733 | 0.635 | 17.481 | 17.783 | 16.925 |
| ExcelFormer | 0.810 | 0.742 | 0.655 | 17.698 | 17.019 | 15.406 |
| TabR | 0.809 | 0.750 | 0.657 | 18.179 | 15.925 | 14.915 |
| DCN-v2 | 0.794 | 0.725 | 0.618 | 18.660 | 18.708 | 18.274 |
| ResNet | 0.800 | 0.728 | 0.641 | 18.717 | 18.981 | 17.594 |
| TANGOS | 0.799 | 0.731 | 0.641 | 19.038 | 18.689 | 17.028 |
| MLP | 0.787 | 0.720 | 0.614 | 20.349 | 19.774 | 20.575 |
| SAINT | 0.791 | 0.726 | 0.623 | 20.594 | 19.406 | 17.953 |
| AutoInt | 0.779 | 0.718 | 0.601 | 22.132 | 21.358 | 21.547 |
| DANets | 0.771 | 0.705 | 0.601 | 22.368 | 21.396 | 21.642 |
| SNN | 0.773 | 0.708 | 0.584 | 22.585 | 22.094 | 22.245 |
| TabTransformer | 0.762 | 0.699 | 0.566 | 22.594 | 21.764 | 22.151 |
| SwitchTab | 0.766 | 0.700 | 0.590 | 23.509 | 23.972 | 22.755 |
| TabCaps | 0.744 | 0.701 | 0.580 | 25.755 | 23.708 | 24.066 |
| NODE | 0.754 | 0.695 | 0.531 | 26.000 | 24.453 | 27.226 |
| TabNet | 0.712 | 0.685 | 0.561 | 29.217 | 26.557 | 26.613 |
| GrowNet | 0.682 | 0.641 | 0.522 | 29.679 | 29.358 | 28.302 |

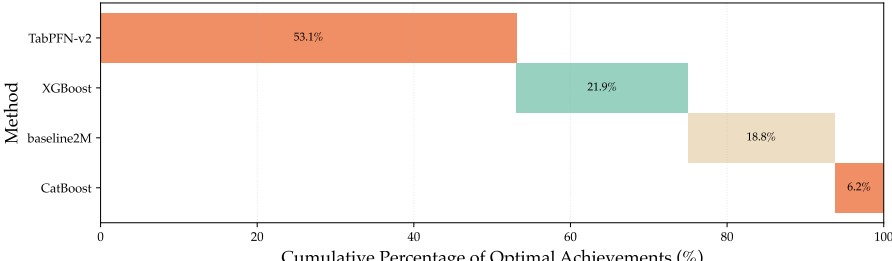

*Figure 6.* Cumulative Percentage of Optimal Achievements of Baseline2M on CTR23.

*Table 16.* Regression results on **BCCO-REG**, sorted by mean $R^2$ in descending order, with the parameter counts of all foundation models highlighted in blue.

| | BCCO-REG | | | |
| --- | --- | --- | --- | --- |
| | Mean | | Rank | |
| Model | $R^2$ (↑) | RMSE (↓) | $R^2$ (↓) | RMSE (↓) |
| LimiX-16M (16.52M) | 0.794 | 0.386 | 3.860 | 4.700 |
| AutoGluon | 0.781 | 0.398 | 5.140 | 5.120 |
| TabM | 0.773 | 0.397 | 5.400 | 5.400 |
| RealMLP | 0.766 | 0.402 | 5.960 | 5.880 |
| TabPFN-v2 (7.24M) | 0.772 | 0.404 | 6.440 | 6.340 |
| LimiX-2M (1.92M) | 0.785 | 0.392 | 6.580 | 6.460 |
| XGBoost | 0.764 | 0.415 | 9.740 | 9.780 |
| LightGBM | 0.715 | 0.423 | 10.060 | 10.040 |
| ET | 0.757 | 0.431 | 12.180 | 12.120 |
| CatBoost | 0.741 | 0.427 | 12.660 | 12.460 |
| RF | 0.752 | 0.438 | 13.460 | 13.440 |
| ExcelFormer | 0.743 | 0.443 | 13.640 | 13.700 |
| T2G-Former | 0.743 | 0.442 | 14.940 | 14.880 |
| FT-Transformer | 0.737 | 0.448 | 15.520 | 15.500 |
| DCN-v2 | 0.739 | 0.448 | 15.840 | 15.840 |
| TabR | 0.733 | 0.448 | 16.580 | 16.660 |
| ResNet | 0.720 | 0.468 | 17.100 | 17.060 |
| ModernNCA | 0.598 | 0.471 | 17.300 | 17.200 |
| Mitra (75.67M) | 0.667 | 0.474 | 17.700 | 17.620 |
| TANGOS | 0.719 | 0.468 | 17.740 | 17.740 |
| MLP-PLR | 0.734 | 0.453 | 17.800 | 17.800 |
| SAINT | 0.701 | 0.481 | 18.540 | 18.560 |
| MLP | 0.701 | 0.487 | 19.020 | 19.080 |
| AutoInt | 0.724 | 0.465 | 19.180 | 19.100 |
| NODE | 0.643 | 0.543 | 22.100 | 22.020 |
| TabNet | 0.670 | 0.516 | 22.720 | 22.660 |
| DNNR | -2.152 | 1.329 | 22.860 | 22.880 |
| SNN | 0.434 | 0.720 | 27.220 | 27.220 |
| GrowNet | 0.201 | 0.864 | 28.880 | 28.860 |
| DANets | 0.005 | 0.979 | 30.480 | 30.500 |
| TabTransformer | -0.000 | 0.981 | 30.640 | 30.660 |
| SwitchTab | 0.001 | 0.981 | 30.720 | 30.720 |

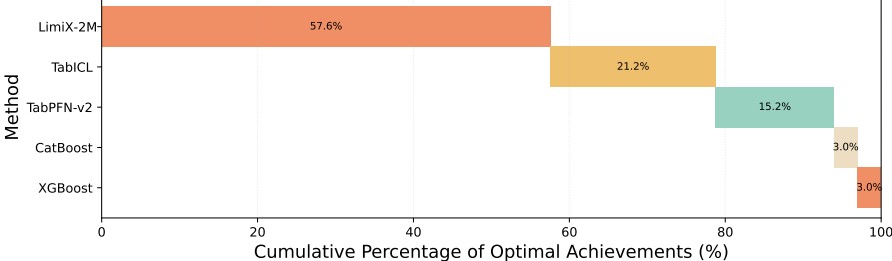

*Figure 7.* Cumulative Percentage of Optimal Achievements of LimiX-2M on CTR23.

*Table 17.* Classification results on **CC18**, sorted by mean AUC in descending order, with the parameter counts of all foundation models highlighted in blue.

| Model | OpenML-cc18 | | | | | |
| | Mean | | | Rank | | |
| | AUC (↑) | Acc. (↑) | F1 (↑) | AUC (↓) | Acc. (↓) | F1 (↓) |
|---|---|---|---|---|---|---|
| LimiX-16M (16.52M) | 0.939 | 0.893 | 0.807 | 4.475 | 5.712 | 4.780 |
| LimiX-2M (1.92M) | 0.935 | 0.892 | 0.799 | 5.475 | 5.695 | 6.322 |
| AutoGluon | 0.931 | 0.885 | 0.785 | 7.254 | 7.814 | 8.305 |
| TabPFN-v2 (7.24M) | 0.929 | 0.887 | 0.786 | 8.593 | 8.305 | 8.780 |
| TabICL (27.10M) | 0.933 | 0.881 | 0.786 | 8.915 | 10.695 | 10.508 |
| TabM | 0.926 | 0.881 | 0.775 | 9.339 | 11.559 | 11.339 |
| RealMLP | 0.925 | 0.883 | 0.789 | 10.712 | 9.153 | 8.220 |
| XGBoost | 0.928 | 0.879 | 0.770 | 10.746 | 11.864 | 11.932 |
| LightGBM | 0.927 | 0.879 | 0.770 | 11.102 | 11.102 | 11.271 |
| CatBoost | 0.926 | 0.870 | 0.765 | 13.322 | 14.729 | 14.746 |
| Mitra (75.67M) | 0.922 | 0.869 | 0.739 | 13.712 | 15.102 | 16.915 |
| TANGOS | 0.914 | 0.868 | 0.758 | 14.915 | 14.814 | 15.000 |
| ExcelFormer | 0.919 | 0.872 | 0.770 | 14.966 | 13.915 | 13.763 |
| RF | 0.925 | 0.872 | 0.757 | 15.034 | 14.305 | 15.339 |
| ET | 0.924 | 0.864 | 0.717 | 15.051 | 17.017 | 19.153 |
| ResNet | 0.917 | 0.865 | 0.765 | 15.102 | 14.983 | 14.068 |
| MLP | 0.913 | 0.861 | 0.742 | 15.508 | 15.712 | 16.797 |
| T2G-Former | 0.912 | 0.864 | 0.748 | 16.864 | 16.407 | 16.203 |
| MLP-PLR | 0.902 | 0.864 | 0.733 | 17.254 | 17.017 | 17.407 |
| ModernNCA | 0.912 | 0.865 | 0.747 | 18.153 | 17.475 | 16.797 |
| TabR | 0.907 | 0.870 | 0.757 | 18.254 | 14.797 | 14.441 |
| FT-Transformer | 0.909 | 0.862 | 0.737 | 18.373 | 16.864 | 17.983 |
| AutoInt | 0.909 | 0.852 | 0.727 | 19.864 | 19.847 | 19.898 |
| DANets | 0.904 | 0.844 | 0.712 | 19.898 | 19.288 | 20.475 |
| DCN-v2 | 0.904 | 0.856 | 0.727 | 20.525 | 19.339 | 19.966 |
| SNN | 0.901 | 0.839 | 0.694 | 22.441 | 22.186 | 22.847 |
| SwitchTab | 0.887 | 0.823 | 0.666 | 23.441 | 22.966 | 23.712 |
| TabCaps | 0.877 | 0.852 | 0.688 | 23.881 | 20.763 | 22.373 |
| TabTransformer | 0.854 | 0.799 | 0.631 | 24.492 | 23.475 | 24.186 |
| NODE | 0.894 | 0.813 | 0.622 | 25.237 | 26.305 | 28.390 |
| SAINT | 0.531 | 0.508 | 0.451 | 26.814 | 26.288 | 24.831 |
| TabNet | 0.869 | 0.822 | 0.664 | 27.390 | 25.034 | 26.780 |
| GrowNet | 0.819 | 0.749 | 0.571 | 27.983 | 28.271 | 27.559 |

*Table 18.* Classification results on **PFN-CLS**, sorted by mean AUC in descending order, with the parameter counts of all foundation models highlighted in blue.

| | PFN-CLS | | | | | |
| | Mean | | | Rank | | |
| Model | AUC (↑) | Acc. (↑) | F1 (↑) | AUC (↓) | Acc. (↓) | F1 (↓) |
|---|---|---|---|---|---|---|
| LimiX-16M (16.52M) | 0.923 | 0.862 | 0.786 | 2.207 | 3.276 | 3.138 |
| LimiX-2M (1.92M) | 0.913 | 0.848 | 0.766 | 4.207 | 6.828 | 6.103 |
| TabPFN-v2 (7.24M) | 0.910 | 0.845 | 0.756 | 5.828 | 7.586 | 7.690 |
| AutoGluon | 0.906 | 0.835 | 0.738 | 6.207 | 8.207 | 8.000 |
| TabICL (27.10M) | 0.903 | 0.832 | 0.742 | 7.414 | 9.241 | 9.724 |
| XGBoost | 0.898 | 0.831 | 0.733 | 9.655 | 8.069 | 9.103 |
| Mitra (75.67M) | 0.897 | 0.826 | 0.719 | 10.069 | 10.828 | 12.655 |
| LightGBM | 0.893 | 0.826 | 0.725 | 10.103 | 9.483 | 10.034 |
| TabM | 0.895 | 0.829 | 0.734 | 10.690 | 10.276 | 10.724 |
| CatBoost | 0.895 | 0.819 | 0.720 | 11.276 | 11.034 | 12.379 |
| RealMLP | 0.889 | 0.823 | 0.732 | 11.655 | 12.897 | 12.448 |
| RF | 0.896 | 0.822 | 0.721 | 12.172 | 11.724 | 12.897 |
| ET | 0.893 | 0.809 | 0.675 | 13.103 | 14.483 | 16.379 |
| ExcelFormer | 0.883 | 0.812 | 0.713 | 15.655 | 13.379 | 14.828 |
| ResNet | 0.869 | 0.801 | 0.700 | 16.414 | 16.586 | 16.000 |
| TANGOS | 0.865 | 0.797 | 0.698 | 17.310 | 18.000 | 16.310 |
| MLP | 0.866 | 0.795 | 0.695 | 18.586 | 17.690 | 17.207 |
| T2G-Former | 0.848 | 0.792 | 0.688 | 19.621 | 18.034 | 17.931 |
| ModernNCA | 0.860 | 0.798 | 0.692 | 19.724 | 17.828 | 16.655 |
| FT-Transformer | 0.849 | 0.789 | 0.683 | 20.414 | 19.931 | 18.621 |
| SwitchTab | 0.858 | 0.776 | 0.626 | 20.586 | 21.517 | 21.103 |
| MLP-PLR | 0.848 | 0.786 | 0.650 | 20.724 | 21.034 | 22.103 |
| DANets | 0.844 | 0.770 | 0.631 | 21.138 | 20.724 | 21.862 |
| TabCaps | 0.834 | 0.788 | 0.636 | 22.448 | 20.276 | 21.448 |
| TabR | 0.842 | 0.789 | 0.688 | 23.034 | 19.655 | 18.241 |
| TabTransformer | 0.821 | 0.761 | 0.604 | 23.517 | 22.621 | 24.414 |
| DCN-v2 | 0.846 | 0.771 | 0.633 | 24.172 | 24.862 | 25.172 |
| NODE | 0.844 | 0.754 | 0.535 | 24.241 | 25.828 | 28.034 |
| AutoInt | 0.838 | 0.772 | 0.640 | 24.345 | 23.172 | 23.069 |
| SAINT | 0.708 | 0.669 | 0.563 | 24.448 | 23.379 | 22.310 |
| SNN | 0.831 | 0.762 | 0.595 | 25.517 | 23.069 | 25.103 |
| TabNet | 0.825 | 0.768 | 0.631 | 27.207 | 24.897 | 26.069 |
| GrowNet | 0.756 | 0.689 | 0.497 | 29.862 | 28.655 | 28.172 |

*Table 19.* Regression results on **PFN-REG**, sorted by mean $R^2$ in descending order, with the parameter counts of all foundation models highlighted in blue.

| | PFN-REG | | | |
| | Mean | | Rank | |
| Model | $R^2$ (↑) | RMSE (↓) | $R^2$ (↓) | RMSE (↓) |
|---|---|---|---|---|
| LimiX-16M (16.52M) | 0.682 | 0.468 | 5.148 | 6.148 |
| LimiX-2M (1.92M) | 0.687 | 0.464 | 5.889 | 5.889 |
| RealMLP | 0.675 | 0.471 | 7.519 | 7.407 |
| AutoGluon | 0.669 | 0.484 | 7.593 | 7.593 |
| TabM | 0.677 | 0.468 | 7.667 | 7.556 |
| TabPFN-v2 (7.24M) | 0.667 | 0.478 | 8.630 | 8.370 |
| XGBoost | 0.661 | 0.493 | 10.481 | 10.481 |
| LightGBM | 0.656 | 0.499 | 11.111 | 11.037 |
| CatBoost | 0.652 | 0.501 | 12.519 | 12.444 |
| ET | 0.643 | 0.520 | 12.630 | 12.556 |
| Mitra (75.67M) | 0.620 | 0.534 | 14.333 | 14.407 |
| RF | 0.634 | 0.529 | 14.889 | 14.963 |
| T2G-Former | 0.630 | 0.507 | 15.037 | 14.926 |
| ExcelFormer | 0.637 | 0.517 | 15.407 | 15.444 |
| FT-Transformer | 0.627 | 0.511 | 15.741 | 15.630 |
| MLP | 0.565 | 0.582 | 15.926 | 16.074 |
| MLP-PLR | 0.629 | 0.516 | 16.074 | 16.148 |
| TabR | 0.628 | 0.511 | 16.148 | 16.000 |
| ModernNCA | 0.636 | 0.512 | 16.222 | 16.407 |
| DCN-v2 | 0.629 | 0.514 | 16.444 | 16.259 |
| ResNet | 0.588 | 0.559 | 16.481 | 16.519 |
| TANGOS | 0.576 | 0.568 | 16.926 | 17.074 |
| SAINT | -8.420 | 0.618 | 17.926 | 17.519 |
| AutoInt | 0.599 | 0.544 | 20.185 | 20.074 |
| NODE | 0.487 | 0.668 | 21.741 | 21.556 |
| TabNet | 0.416 | 0.647 | 22.963 | 22.889 |
| DNNR | -8.173 | 2.289 | 24.222 | 24.370 |
| SNN | 0.364 | 0.764 | 26.185 | 26.222 |
| GrowNet | 0.087 | 0.939 | 27.556 | 27.519 |
| DANets | 0.001 | 0.988 | 28.815 | 28.852 |
| SwitchTab | -0.025 | 1.001 | 29.259 | 29.296 |
| TabTransformer | -0.020 | 0.998 | 30.333 | 30.370 |

*Table 20.* Classification results on **TabArena-CLS**, sorted by mean AUC in descending order, with the parameter counts of all foundation models highlighted in blue.

| | TabArena | | | | | |
| | Mean | | | Rank | | |
| Model | AUC (↑) | Acc. (↑) | F1 (↑) | AUC (↓) | Acc. (↓) | F1 (↓) |
|---|---|---|---|---|---|---|
| LimiX-16M (16.52M) | 0.849 | 0.877 | 0.597 | 3.636 | 3.424 | 7.273 |
| LimiX-2M (1.92M) | 0.846 | 0.876 | 0.594 | 5.000 | 3.394 | 6.909 |
| AutoGluon | 0.844 | 0.870 | 0.574 | 5.909 | 5.606 | 9.545 |
| TabICL (27.10M) | 0.840 | 0.870 | 0.553 | 7.182 | 6.636 | 11.000 |
| TabPFN-v2 (7.24M) | 0.838 | 0.872 | 0.589 | 7.485 | 4.697 | 9.152 |
| LightGBM | 0.841 | 0.868 | 0.574 | 7.606 | 8.606 | 10.970 |
| XGBoost | 0.838 | 0.867 | 0.567 | 8.545 | 7.970 | 11.273 |
| RF | 0.837 | 0.864 | 0.558 | 10.061 | 8.697 | 12.545 |
| CatBoost | 0.835 | 0.867 | 0.574 | 10.273 | 7.818 | 10.242 |
| ET | 0.833 | 0.857 | 0.505 | 11.212 | 11.515 | 16.879 |
| Mitra (75.67M) | 0.815 | 0.862 | 0.533 | 12.667 | 10.636 | 15.545 |
| TabM | 0.807 | 0.855 | 0.516 | 15.212 | 15.212 | 13.970 |
| ExcelFormer | 0.810 | 0.849 | 0.555 | 15.455 | 17.485 | 15.515 |
| RealMLP | 0.809 | 0.751 | 0.477 | 17.303 | 25.121 | 18.061 |
| MLP | 0.772 | 0.822 | 0.459 | 19.212 | 18.212 | 20.212 |
| TANGOS | 0.791 | 0.844 | 0.522 | 19.455 | 18.364 | 16.576 |
| T2G-Former | 0.779 | 0.822 | 0.482 | 19.697 | 20.576 | 16.273 |
| MLP-PLR | 0.781 | 0.836 | 0.460 | 19.818 | 19.485 | 20.152 |
| ModernNCA | 0.783 | 0.846 | 0.511 | 20.576 | 21.061 | 17.667 |
| AutoInt | 0.769 | 0.826 | 0.474 | 20.636 | 21.121 | 19.182 |
| TabR | 0.785 | 0.842 | 0.510 | 21.061 | 20.182 | 16.545 |
| NODE | 0.769 | 0.792 | 0.352 | 21.182 | 21.939 | 24.970 |
| ResNet | 0.781 | 0.824 | 0.532 | 21.364 | 22.182 | 16.818 |
| FT-Transformer | 0.770 | 0.803 | 0.468 | 21.485 | 22.061 | 19.030 |
| TabTransformer | 0.739 | 0.781 | 0.438 | 21.667 | 21.970 | 19.606 |
| DCN-v2 | 0.769 | 0.833 | 0.482 | 22.333 | 19.909 | 18.606 |
| SNN | 0.755 | 0.818 | 0.442 | 23.242 | 22.545 | 22.545 |
| TabCaps | 0.742 | 0.837 | 0.471 | 23.273 | 18.515 | 20.212 |
| SwitchTab | 0.754 | 0.799 | 0.409 | 24.091 | 25.545 | 22.758 |
| DANets | 0.749 | 0.776 | 0.453 | 24.273 | 24.303 | 18.727 |
| SAINT | 0.694 | 0.739 | 0.437 | 25.485 | 25.758 | 22.121 |
| TabNet | 0.709 | 0.789 | 0.438 | 26.818 | 22.879 | 23.727 |
| GrowNet | 0.646 | 0.674 | 0.361 | 27.697 | 28.939 | 23.909 |

*Table 21.* Regression results on **TabArena-REG**, sorted by mean $R^2$ in descending order, with the parameter counts of all foundation models highlighted in blue.

| | TabArena-REG | | | |
| | Mean | | Rank | |
| Model | $R^2$ ($\uparrow$) | RMSE ($\downarrow$) | $R^2$ ($\downarrow$) | RMSE ($\downarrow$) |
|---|---|---|---|---|
| AutoGluon | 0.791 | 0.414 | 3.462 | 3.462 |
| LimiX-16M (16.52M) | 0.796 | 0.406 | 3.462 | 3.462 |
| LimiX-2M (1.92M) | 0.788 | 0.413 | 4.538 | 4.538 |
| TabPFN-v2 (7.24M) | 0.777 | 0.422 | 5.923 | 5.923 |
| TabM | 0.777 | 0.424 | 6.692 | 6.692 |
| CatBoost | 0.774 | 0.431 | 7.308 | 7.308 |
| XGBoost | 0.778 | 0.430 | 7.462 | 7.462 |
| RealMLP | 0.776 | 0.426 | 8.000 | 8.000 |
| LightGBM | 0.771 | 0.435 | 8.000 | 8.000 |
| RF | 0.758 | 0.456 | 11.000 | 11.000 |
| ET | 0.746 | 0.464 | 11.385 | 11.308 |
| TabR | 0.729 | 0.476 | 15.231 | 15.231 |
| ExcelFormer | 0.681 | 0.521 | 15.385 | 15.385 |
| NODE | 0.665 | 0.542 | 15.692 | 15.692 |
| TANGOS | 0.673 | 0.534 | 16.538 | 16.538 |
| DCN-v2 | 0.718 | 0.486 | 16.692 | 16.769 |
| Mitra (75.67M) | 0.666 | 0.538 | 17.692 | 17.692 |
| ModernNCA | 0.712 | 0.496 | 18.462 | 18.462 |
| MLP-PLR | 0.714 | 0.496 | 18.538 | 18.538 |
| AutoInt | 0.705 | 0.503 | 19.154 | 19.154 |
| MLP | 0.694 | 0.516 | 19.615 | 19.615 |
| ResNet | 0.687 | 0.521 | 19.769 | 19.769 |
| SAINT | 0.712 | 0.498 | 20.250 | 20.250 |
| T2G-Former | 0.677 | 0.526 | 20.538 | 20.538 |
| TabNet | 0.641 | 0.564 | 20.769 | 20.769 |
| FT-Transformer | 0.626 | 0.572 | 23.077 | 23.077 |
| DNNR | -0.368 | 1.058 | 25.615 | 25.615 |
| SNN | 0.451 | 0.729 | 26.692 | 26.692 |
| GrowNet | 0.316 | 0.814 | 28.385 | 28.385 |
| TabTransformer | 0.016 | 0.993 | 30.333 | 30.333 |
| DANets | 0.012 | 0.998 | 30.462 | 30.462 |
| SwitchTab | 0.004 | 1.002 | 30.923 | 30.923 |

*Table 22.* Classification results on **TabZilla**.

| | TabZilla | | | | | |
| | Mean | | | Rank | | |
| Model | AUC (↑) | Acc. (↑) | F1 (↑) | AUC (↓) | Acc. (↓) | F1 (↓) |
|---|---|---|---|---|---|---|
| LimiX-16M (16.52M) | 0.943 | 0.885 | 0.836 | 5.037 | 5.333 | 6.519 |
| LimiX-2M (1.92M) | 0.938 | 0.883 | 0.832 | 5.963 | 6.704 | 7.074 |
| AutoGluon | 0.933 | 0.871 | 0.803 | 7.889 | 8.259 | 9.704 |
| TabPFN-v2 (7.24M) | 0.929 | 0.863 | 0.797 | 8.704 | 9.815 | 10.185 |
| TabICL (27.10M) | 0.933 | 0.864 | 0.803 | 9.704 | 10.556 | 11.444 |
| XGBoost | 0.929 | 0.863 | 0.789 | 10.111 | 11.407 | 12.778 |
| TabM | 0.928 | 0.869 | 0.816 | 10.148 | 9.889 | 10.407 |
| LightGBM | 0.927 | 0.863 | 0.796 | 11.963 | 11.074 | 11.556 |
| RF | 0.924 | 0.852 | 0.773 | 12.444 | 13.519 | 14.037 |
| RealMLP | 0.923 | 0.872 | 0.815 | 12.481 | 9.185 | 9.333 |
| CatBoost | 0.922 | 0.848 | 0.780 | 13.778 | 14.889 | 14.852 |
| Mitra (75.67M) | 0.915 | 0.841 | 0.758 | 15.815 | 15.667 | 16.593 |
| ExcelFormer | 0.915 | 0.861 | 0.802 | 15.852 | 13.481 | 13.481 |
| T2G-Former | 0.909 | 0.852 | 0.790 | 15.926 | 16.000 | 16.222 |
| ModernNCA | 0.907 | 0.850 | 0.794 | 16.000 | 16.593 | 15.778 |
| ET | 0.912 | 0.837 | 0.745 | 16.370 | 16.704 | 18.000 |
| TabR | 0.904 | 0.853 | 0.793 | 16.852 | 15.074 | 15.481 |
| MLP-PLR | 0.906 | 0.847 | 0.773 | 16.889 | 16.519 | 17.630 |
| TANGOS | 0.909 | 0.841 | 0.776 | 17.000 | 15.519 | 15.963 |
| FT-Transformer | 0.903 | 0.842 | 0.769 | 17.778 | 17.593 | 17.111 |
| MLP | 0.903 | 0.825 | 0.747 | 18.111 | 18.407 | 18.889 |
| ResNet | 0.908 | 0.834 | 0.769 | 18.407 | 17.037 | 16.519 |
| AutoInt | 0.896 | 0.833 | 0.748 | 18.778 | 17.963 | 18.407 |
| DCN-v2 | 0.904 | 0.844 | 0.781 | 19.481 | 18.407 | 18.889 |
| SAINT | 0.824 | 0.764 | 0.680 | 21.037 | 20.778 | 20.444 |
| SNN | 0.874 | 0.816 | 0.706 | 22.519 | 20.556 | 21.778 |
| DANets | 0.881 | 0.800 | 0.712 | 22.926 | 22.000 | 22.519 |
| TabTransformer | 0.814 | 0.759 | 0.659 | 23.296 | 21.630 | 22.593 |
| SwitchTab | 0.860 | 0.764 | 0.660 | 23.741 | 25.519 | 25.111 |
| TabCaps | 0.887 | 0.816 | 0.729 | 25.296 | 22.481 | 22.815 |
| NODE | 0.869 | 0.784 | 0.633 | 25.593 | 25.593 | 27.630 |
| GrowNet | 0.829 | 0.732 | 0.651 | 27.630 | 27.222 | 26.148 |
| TabNet | 0.860 | 0.771 | 0.668 | 28.593 | 28.000 | 27.556 |

*Table 23.* Classification results on **TALENT-CLS**, sorted by mean AUC in descending order, with the parameter counts of all foundation models highlighted in blue.

| | TALENT-CLS | | | | | |
| Model | Mean | | | Rank | | |
| | AUC (↑) | Acc. (↑) | F1 (↑) | AUC (↓) | Acc. (↓) | F1 (↓) |
|---|---|---|---|---|---|---|
| LimiX-16M (16.52M) | 0.903 | 0.861 | 0.752 | 4.212 | 3.380 | 4.464 |
| LimiX-2M (1.92M) | 0.897 | 0.853 | 0.734 | 6.067 | 6.006 | 7.525 |
| TabICL (27.10M) | 0.894 | 0.845 | 0.715 | 6.531 | 7.620 | 9.162 |
| TabPFN-v2 (7.24M) | 0.895 | 0.850 | 0.727 | 7.017 | 6.933 | 8.872 |
| AutoGluon | 0.891 | 0.845 | 0.719 | 7.285 | 7.911 | 8.732 |
| XGBoost | 0.881 | 0.837 | 0.713 | 10.782 | 10.955 | 11.285 |
| TabM | 0.881 | 0.842 | 0.719 | 10.961 | 10.486 | 10.257 |
| LightGBM | 0.880 | 0.836 | 0.713 | 11.117 | 11.302 | 11.609 |
| RealMLP | 0.881 | 0.843 | 0.726 | 11.749 | 9.693 | 9.067 |
| Mitra (75.67M) | 0.882 | 0.834 | 0.689 | 11.899 | 12.743 | 14.810 |
| CatBoost | 0.876 | 0.828 | 0.704 | 13.184 | 13.279 | 13.475 |
| RF | 0.877 | 0.828 | 0.691 | 14.101 | 14.676 | 15.765 |
| ET | 0.875 | 0.821 | 0.662 | 14.916 | 17.380 | 18.944 |
| ExcelFormer | 0.870 | 0.826 | 0.699 | 15.441 | 16.128 | 14.911 |
| ResNet | 0.866 | 0.825 | 0.695 | 16.944 | 16.693 | 14.994 |
| FT-Transformer | 0.859 | 0.822 | 0.678 | 17.771 | 17.939 | 17.542 |
| T2G-Former | 0.858 | 0.823 | 0.683 | 17.877 | 17.559 | 17.056 |
| TANGOS | 0.861 | 0.818 | 0.684 | 18.123 | 18.039 | 16.637 |
| MLP | 0.862 | 0.817 | 0.675 | 18.514 | 18.156 | 18.240 |
| ModernNCA | 0.861 | 0.825 | 0.683 | 19.112 | 18.732 | 17.972 |
| TabR | 0.858 | 0.824 | 0.680 | 19.385 | 17.866 | 16.978 |
| DCN-v2 | 0.854 | 0.815 | 0.662 | 19.894 | 19.978 | 19.715 |
| MLP-PLR | 0.849 | 0.816 | 0.663 | 20.603 | 19.223 | 19.179 |
| DANets | 0.848 | 0.805 | 0.654 | 21.251 | 21.000 | 20.559 |
| SAINT | 0.813 | 0.781 | 0.630 | 22.385 | 21.413 | 20.816 |
| AutoInt | 0.842 | 0.803 | 0.646 | 22.743 | 23.274 | 22.620 |
| SwitchTab | 0.842 | 0.795 | 0.637 | 23.480 | 23.972 | 23.123 |
| TabCaps | 0.834 | 0.813 | 0.654 | 23.536 | 20.313 | 20.737 |
| TabTransformer | 0.832 | 0.790 | 0.627 | 23.827 | 23.726 | 22.821 |
| SNN | 0.836 | 0.796 | 0.625 | 24.162 | 23.972 | 24.011 |
| NODE | 0.830 | 0.779 | 0.570 | 25.022 | 25.296 | 26.754 |
| TabNet | 0.818 | 0.794 | 0.630 | 27.285 | 24.765 | 25.173 |
| GrowNet | 0.743 | 0.704 | 0.542 | 29.553 | 28.978 | 26.905 |

*Table 24.* Regression results on **TALENT-REG**, sorted by mean $R^2$ in descending order, with the parameter counts of all foundation models highlighted in blue.

| | TALENT-REG | | | |
| | Mean | | Rank | |
| Model | $R^2$ (↑) | RMSE (↓) | $R^2$ (↓) | RMSE (↓) |
|---|---|---|---|---|
| LimiX-16M (16.52M) | 0.735 | 0.433 | 3.232 | 3.919 |
| AutoGluon | 0.722 | 0.448 | 5.667 | 5.889 |
| TabM | 0.708 | 0.459 | 6.525 | 6.364 |
| TabPFN-v2 (7.24M) | 0.695 | 0.465 | 7.061 | 6.980 |
| LimiX-2M (1.92M) | 0.721 | 0.451 | 7.061 | 6.980 |
| RealMLP | 0.697 | 0.465 | 7.202 | 7.030 |
| XGBoost | 0.710 | 0.462 | 8.434 | 8.384 |
| LightGBM | 0.707 | 0.461 | 9.283 | 9.253 |
| ET | 0.696 | 0.476 | 10.990 | 10.939 |
| CatBoost | 0.700 | 0.471 | 11.525 | 11.465 |
| RF | 0.697 | 0.474 | 11.596 | 11.596 |
| ExcelFormer | 0.653 | 0.512 | 14.687 | 14.727 |
| T2G-Former | 0.656 | 0.512 | 15.384 | 15.323 |
| TabR | 0.651 | 0.516 | 16.465 | 16.434 |
| DCN-v2 | -0.361 | 0.818 | 16.677 | 16.727 |
| Mitra (75.67M) | 0.602 | 0.547 | 16.889 | 16.909 |
| FT-Transformer | 0.648 | 0.519 | 16.960 | 16.909 |
| MLP-PLR | 0.653 | 0.521 | 17.222 | 17.182 |
| ModernNCA | 0.633 | 0.530 | 17.747 | 17.727 |
| ResNet | 0.562 | 0.550 | 17.879 | 17.848 |
| TANGOS | 0.592 | 0.547 | 18.384 | 18.364 |
| MLP | 0.556 | 0.564 | 18.768 | 18.808 |
| SAINT | -1.541 | 0.571 | 19.515 | 19.384 |
| AutoInt | 0.636 | 0.538 | 20.172 | 20.182 |
| NODE | 0.568 | 0.600 | 21.697 | 21.545 |
| TabNet | 0.576 | 0.586 | 22.869 | 22.818 |
| DNNR | -9.172 | 2.528 | 24.525 | 24.566 |
| SNN | 0.344 | 0.777 | 26.505 | 26.525 |
| GrowNet | -0.182 | 0.920 | 28.121 | 28.152 |
| DANets | 0.005 | 0.998 | 29.434 | 29.455 |
| TabTransformer | 0.001 | 1.001 | 29.576 | 29.626 |
| SwitchTab | -0.002 | 1.002 | 29.939 | 29.980 |

*Table 25.* Regression results on **CTR23**, sorted by mean $R^2$ in descending order, with the parameter counts of all foundation models highlighted in blue.

| | CTR23 | | | |
|---|---|---|---|---|
| | Mean | | Rank | |
| Model | $R^2$ ($\uparrow$) | RMSE ($\downarrow$) | $R^2$ ($\downarrow$) | RMSE ($\downarrow$) |
| LimiX-16M (16.52M) | 0.745 | 0.477 | 4.545 | 4.667 |
| AutoGluon | 0.725 | 0.497 | 5.939 | 5.848 |
| RealMLP | 0.721 | 0.494 | 6.333 | 6.303 |
| TabM | 0.719 | 0.494 | 6.515 | 6.545 |
| LimiX-2M (1.92M) | 0.730 | 0.495 | 7.636 | 7.667 |
| TabPFN-v2 (7.24M) | 0.716 | 0.503 | 8.485 | 8.394 |
| XGBoost | 0.712 | 0.511 | 9.818 | 9.879 |
| LightGBM | 0.706 | 0.516 | 10.848 | 10.939 |
| ET | 0.697 | 0.535 | 11.939 | 12.000 |
| CatBoost | 0.700 | 0.528 | 12.394 | 12.394 |
| RF | 0.694 | 0.539 | 12.818 | 12.788 |
| ExcelFormer | 0.665 | 0.556 | 13.848 | 13.939 |
| T2G-Former | 0.674 | 0.544 | 15.061 | 15.030 |
| DCN-v2 | 0.670 | 0.545 | 15.788 | 15.848 |
| ResNet | 0.645 | 0.587 | 15.879 | 15.939 |
| FT-Transformer | 0.667 | 0.549 | 15.909 | 16.000 |
| TabR | 0.671 | 0.543 | 16.182 | 16.091 |
| MLP-PLR | 0.672 | 0.553 | 16.909 | 16.939 |
| ModernNCA | 0.667 | 0.550 | 17.091 | 16.788 |
| MLP | 0.608 | 0.623 | 17.121 | 17.121 |
| SAINT | 0.654 | 0.561 | 17.939 | 17.848 |
| Mitra (75.67M) | 0.624 | 0.583 | 17.939 | 18.061 |
| TANGOS | 0.642 | 0.586 | 18.121 | 18.152 |
| AutoInt | 0.655 | 0.568 | 18.970 | 18.939 |
| NODE | 0.568 | 0.666 | 21.697 | 21.545 |
| TabNet | 0.605 | 0.623 | 21.818 | 21.879 |
| DNNR | -2.969 | 1.651 | 23.909 | 23.909 |
| SNN | 0.369 | 0.834 | 26.758 | 26.788 |
| GrowNet | 0.185 | 0.944 | 28.667 | 28.636 |
| DANets | 0.001 | 1.052 | 30.000 | 30.000 |
| TabTransformer | 0.000 | 1.053 | 30.030 | 30.030 |
| SwitchTab | -0.006 | 1.057 | 31.091 | 31.091 |

*Table 26.* Inference time comparison in milliseconds (ms). The experiments were conducted using an **AMD EPYC 9354 32-Core Processor** for CPU evaluation and an **NVIDIA GeForce RTX 4090** for GPU evaluation. The best results are highlighted in **bold**.

| Model | CPU (ms) | GPU (ms) |
|---|---|---|
| TabPFN-v2 | 51950.08 | 352.60 |
| LimiX-16M | 68447.99 | 368.08 |
| TabICL | 22161.85 | 1749.61 |
| Mitra | 124453.05 | 5766.25 |
| LimiX-2M | **17257.34** | **171.40** |

