# OpenReview forum: "LimiX-2M: Mitigating Low-Rank Collapse and Attention Bottlenecks in Tabular Foundation Models"
_ICML.cc/2026/Conference — ICML 2026 regular_

### Official Review · Reviewer_qq5Z · 2026-03-04

**Soundness:** 3
**Presentation:** 1
**Significance:** 3
**Originality:** 3
**Overall Recommendation:** 5
**Confidence:** 4

**Summary:**

This paper proposes **MiniX**, a compact tabular foundation model built around two changes: (i) **RaBEL**, a radial-basis embedding layer that expands each numeric scalar into a small bank of localized RBF features with exponent-gating (ii) a **reordered bidirectional attention block** from the common feature→sample→FFN pattern to **sample-attention → FFN → feature-attention (SNF)**, paired with attention pooling so that the attention computations influence the final prediction. The motivation is that standard linear scalar tokenization yields low effective rank and weak early-layer value sensitivity, causing representational redundancy and inefficient scaling. The paper provides theoretical and empirical evidence of this low-rank behavior and reports benchmark results where MiniX is claimed to outperform or be competitive with larger tabular FMs (notably TabPFN-v2 and TabICL) across suites like OpenML-CC18, TALENT, TabArena, TabZilla, and BCCO.

**Compliance With Llm Reviewing Policy:**

Affirmed.

**Final Justification:**

The authors have addressed all my concerns during the rebuttal period. Therefore, I am increasing my score and recommending acceptance.

**Note:** I still believe the overall organisation and clarity of the flow in the paper needs serious improvement. Since the authors were not able to modify the manuscript during the rebuttal, I am counting on their good faith to carefully integrate the requested changes in the final camera-ready version.

**Key Questions For Authors:**

1. **Evaluation protocol:** Are all foundation models compared in the large benchmark table evaluated under the *same* constraints (context length, number of in-context training points, preprocessing, ensembling, calibration), and are the exact checkpoints and inference settings public/reproducible? If the results are taken from the previous work, this needs to be highlighted, while also making sure that the evaluation protocols are satisified.
2. **Compute accounting:** What are the actual wall-clock and memory costs (training + inference) vs TabPFN-v2 at matched accuracy—especially considering RaBEL’s RBF expansion and exponent gating overhead?
3. **Tokenizer hyperparameters:** How sensitive are results to the number of RBF centers, initialization via quantiles/IQR, and the exponent-bin design? Is there a stable “default” across suites?

**Strengths And Weaknesses:**

### Strengths

* **Clear diagnosis + mechanism-driven fix.** The low-rank collapse argument is supported both theoretically (rank bounds under linear embedding) and empirically (rank metrics across layers; low-rank truncation barely hurting TabPFN-v2).
* **Simple architectural interventions.** Both RaBEL and SNF are relatively surgical: swap tokenizer and reorder modules without adding many new moving parts, which plausibly explains the parameter-efficiency gains.
* **Evidence that RaBEL helps beyond MLP backbones.** They compare embedding methods in an MLP setting and then within a small transformer FM training setup; RaBEL tends to improve over MLP/PLE/Periodic in those reported tables.

### Weaknesses

* **Motivation behind the exponent-gating is lacking.** The choice to introduce an MLP to modulate scale is unintuitive to me. Since the columns are standardized before the embedding, I would not expect the scale to matter as much. A toy example or a theoretical analysis would help to elucidate this point. Otherwise, it feels like an unnecessary moving part.
* **Questionable baseline tuning.** The reported results for PLE and PLR embeddings of Gorishnyi (2022) in Table 2 raise concerns. For example on GC and UK datasets, they fall behind the MLP with no embeddings. I think this merits further exploration. I suspect something is off in the dataset pre-processing pipeline.
* **Benchmarking clarity / apples-to-apples risk.** “Aggregated rank” tables across many suites are hard to interpret without stronger normalization of training budgets, pretraining corpora, and evaluation protocols across models. It would be great to include per-dataset tables similar to what has been done in TabArena and TabM papers.
* **Attribution between RaBEL vs SNF vs readout changes.** The narrative emphasizes two components, but the readout detail (“attention pooling over all feature tokens”) is also a meaningful change; it’s not fully isolated as a separate factor in the core claims about “all attention contributes.”
* **Bad clarity and presentation.** Section 5 of the paper is poorly organized. For example, useful space is wasted in Table 2 for reporting 2 metrics, when 1 metric / aggregate would suffice. Figure 2 is **wrong**: attention weights in (b) for the FNS appear more diverse than the ones in (c) for the new SNF -- I suspect this to be a typo. Section 5.5.2 is too short and useless. Tables 3/4 are not described in the text. Table 6 taking the whole page in the main text is egregious.

---

> ### Author Rebuttal · Authors · 2026-03-31
>
> We thank the reviewer for the recognition of our work and for the valuable comments. We briefly address the concerns below.
>
> # Weakness
>
> > **Q1: Motivation of exponent-gating.**
>
> A1: We consider the following scenario: some feature values lie very close to the mean, so after normalization their magnitudes become extremely small. In this case, the model may tend to treat them as zeros. To better capture such subtle variations, we introduce exponent-gating, which separates the mantissa and exponent of each value. This can be viewed as a form of rescaling that helps the model remain sensitive to fine-grained differences in small-value regions. To verify this effect, we designed a toy experiment using different random seeds to construct features that become very small after normalization. We then compared RaBEL without gating against RaBEL with gating. The experimental setup and results are available at https://anonymous.4open.science/r/toy-ablation-experiment-of-RaBEL-gating-52BF.
>
> > **Q2: GC and UK datasets results for PLE and PLR.**
>
> A2: We carefully rechecked the data preprocessing and reran PLE and PLR on UK and GC five times. The new results show that the previously reported numbers fall within the normal range of variation. We also observe that, on UK and MA, MLP-PLE, MLP-Periodic, and MLP-RaBEL all perform worse than the plain MLP baseline. This suggests that, although numerical embeddings are effective for most features, they may not be suitable for all numerical features. The underlying mechanism still requires further investigation.
>
> > **Q3: Benchmarking clarity and per-dataset tables.**
>
> A3: Rank-based comparison is a standard and widely used practice in tabular foundation models, as adopted in, e.g.[1][2], We use **aggregated rank** to measure overall average performance, although we agree that it is less fine-grained. Related works also use summary statistics at the dataset level: TabM reports dataset-level mean rank and mean score, while TabArena uses dataset-level Elo. In our paper, Appendix Tables 14–24 (Lines 834–1420) provide dataset-level mean rank and mean score across 11 benchmarks, using AUC/Acc/F1 for classification and $RMSE/(R^2)$ for regression. We therefore believe that our evaluation has a granularity comparable to prior work while covering a broad set of benchmarks.
>
> > **Q4: About readout change.**
>
> A4: What we want to convey is that, by reordering the attention blocks, the final feature-attention layer can leverage the (p(y \mid x)) relationship modeled in the preceding layers to infer the output, thereby affecting the readout. In this sense, the change in readout is a consequence of the attention reordering, rather than a separate factor.
>
> > **Q5: Organization of Section 5 and Figure 2.**
>
> Thank you for the helpful suggestions on the presentation of Section 5. We agree that Table 2 could be better formatted. As for Section 5.5.2, its purpose is to direct readers to the ablation results in the appendix rather than to serve as a standalone substantive section, and we hope this clarification is helpful.
> Regarding Fig. 2, thank you very much for pointing this out. There was indeed a formatting error on our side. We have corrected Fig. 2 and provide the revised version in: https://anonymous.4open.science/r/Revised-Figure-2/.
>
> # Key Questions
> > **KQ1: About Evaluation protocol.**
>
> A1: All foundation models were evaluated using their standard official settings. We unified the context length, ensemble count, and preprocessing pipeline across methods, and all evaluation results are fully reproducible.
>
> > **KQ2: About Compute accounting.**
>
> A2: We do not have access to the training time or memory usage of TabPFN-v2. Instead, we compare inference time on the same hardware. The results are reported in Table 25 (Lines 1444–1453).
>
> > **KQ3：RaBEL hyperparameters.**
>
> A3: We provide an ablation study of RaBEL’s hyperparameters in Appendix C.2 (Lines 781–810). The best setting is highlighted in bold, and this is also the default configuration we use.
>
> **[1]** Ye, Han-Jia, et al. "A closer look at deep learning methods on tabular datasets." arXiv preprint arXiv:2407.00956 (2024).
>
> **[2]** Qu, Jingang, et al. "Tabicl: A tabular foundation model for in-context learning on large data." arXiv preprint arXiv:2502.05564 (2025).

---

> > ### Author Rebuttal · Reviewer_qq5Z · 2026-04-01
> >
> > I thank the authors for the rebuttal and the additional clarifications. The responses were helpful, and I am persuaded that the core idea is sound. I am therefore willing to raise my score, if the authors close the loop on the following issues:
> >
> > - The attribution between SNF and the changed readout is still somewhat entangled. Do you have an ablation that shows that pooling works better than using the target token alone?
> > - The baseline-comparison issue for PLE/PLR deserves further exploration. In the manuscript, PLE and Periodic are not used in their original form; the authors explicitly modify them to use shared linear projections instead of feature-specific ones to fit the FM setup. Can you re-run the MLP experiment on the UK and GC datasets with untied linear embedding layer, could this be the reason for under-performance?
> > - Finally, I also strongly encourage the authors to substantially improve the presentation in the final version if the paper gets accepted.
> >
> > With these revisions, I would be comfortable supporting acceptance more strongly.

---

> > > ### Author Response · Authors · 2026-04-01
> > >
> > > > **Q1: Clarifying the role of the target-token readout in SNF**
> > >
> > > **A1:** We appreciate the reviewer’s concern that the effect of SNF may appear confounded with the readout design.
> > >
> > > To clarify, in our paper, *attention pooling* does **not** refer to an additional pooling module appended to the model, nor do we introduce a separate readout mechanism that replaces the target token. Our objective remains standard in-context prediction of y = f(x).
> > >
> > > Under the SNF ordering, each layer first applies sample attention to aggregate column-wise statistics and cross-sample regularities, followed by a lightweight feed-forward transformation, and then feature attention to model inter-feature dependencies. Crucially, after the final feature-attention layer, the loss is computed **only on the target token**. Consequently, in the last feature-attention block, the target token attends to and aggregates information from all feature tokens through its attention weights. In this sense, the “attention pooling” we referred to is simply an **implicit effect of the final feature-attention operation**, not an additional architectural component.
> > >
> > > Therefore, our model does not perform any extra attention-pooling step beyond the standard target-token readout. Rather, the target token in the last feature-attention layer already functions as an attention-based aggregation over feature tokens. We agree that our original wording may have obscured this distinction, and we will revise the manuscript to make this point explicit.
> > >
> > > ---
> > >
> > > > **Q2: Revisiting PLE/PLR with feature-specific embedding layers**
> > >
> > > **A2:** Thank you for this valuable suggestion. We agree that the comparison involving PLE/PLR deserves further examination.
> > >
> > > Our original goal was to study embedding designs under the **tabular foundation model** setting. Accordingly, all embedding variants in the manuscript were evaluated under the same FM-style setup, which uses **shared** linear projections rather than **feature-specific** ones. As the reviewer noted, this differs from the original formulation of PLE/Periodic and could affect their performance.
> > >
> > > Following this suggestion, we re-ran the MLP experiments on the **UK** and **GC** datasets using **untied, feature-specific linear embedding layers**, more closely matching the original design of these methods. We indeed observe a substantial improvement under this setting, and we report the new results below.
> > >
> > > We found that adding feature-specific linear layers leads to a substantial improvement for PLE, while the other embedding methods do not show comparable gains under the same modification.
> > > These observations suggest that PLE may be better suited to conventional tabular deep models, but may be less compatible with the embedding design requirements of the tabular foundation model paradigm. That said, we view these findings as preliminary, and further experiments will be needed to draw a more definitive conclusion.
> > >
> > > | **AUC / \(R^2\)** | GC($\uparrow$)   | UK($\uparrow$)  |
> > > |-------------------|-------|-------|
> > > | MLP               | 0.696 | 0.984 |
> > > | MLP-MLP           | 0.905 | 0.985 |
> > > | MLP-PLE           | 0.972 | 0.968 |
> > > | MLP-Periodic      | 0.621 | 0.979 |
> > > | MLP-RaBEL         | 0.974 | 0.977 |
> > >
> > > | **ACC / RMSE**    | GC($\uparrow$)   | UK($\uparrow$) |
> > > |-------------------|-------|-------|
> > > | MLP               | 0.324 | 0.867 |
> > > | MLP-MLP           | 0.652 | 0.917 |
> > > | MLP-PLE           | 0.809 | 0.892 |
> > > | MLP-Periodic      | 0.292 | 0.910 |
> > > | MLP-RaBEL         | 0.787 | 0.884 |
> > >
> > > ---
> > >
> > > > **Q3: Presentation and writing quality**
> > >
> > > **A3:** We appreciate this comment and agree that the paper presentation can be further improved. If accepted, we will carefully revise the writing, organization, and overall clarity of the manuscript in the final version.

---

### Official Review · Reviewer_eVJd · 2026-03-05

**Soundness:** 3
**Presentation:** 2
**Significance:** 3
**Originality:** 3
**Overall Recommendation:** 5
**Confidence:** 4

**Summary:**

The paper introduces MiniX, a small tabular foundation model (2M parameters) with two architectural improvements: A new numerical embedding module (RaBEL) based on radial basis functions, and a reordering of the attention and feed-forward layers. Ablations and experiments show the benefit of the proposed improvements and give some intuition, and benchmarks show that MiniX achieves strong performance, not as good as the best model but it is much smaller and cheaper.

**Compliance With Llm Reviewing Policy:**

Affirmed.

**Final Justification:**

The paper introduces a new efficient tabular foundation model with two main improvements whose benefit is shown in ablation studies. The reported performance, if it holds up, is impressive, and especially so given the short pre-training time. My concerns regarding the evaluation have been resolved in the rebuttal, and the authors provide even more ablations. My concerns regarding the reproducibility are not fully resolved, as the authors mention some pre-training details but do not provide any details about the pre-training data generation. I am thus not fully decided between giving a 4 or 5, but I think the paper will be interesting to the community regardless and recommend acceptance.

**Key Questions For Authors:**

(Q1) A 12-layer FSN (TabPFN-style) network contains an 11-layer SNF network: (FSN)^{12} = F(SNF)^{11}SN. I would expect an 11-layer SNF network to be not much worse than a 12-layer SNF network, in which case the question to me is: Why would the extra layers in a 12-layer FSN make it worse than an 11-layer SNF? Regarding Figure 2, I would suspect that the first F layer in FSN mainly diffuses information from the target to the other features, which would not be shown in the figure. The second F layer in FSN should then do something similar to the first F layer in SNF. (These are just some thoughts, I don't know what should be done with them. But it could at least be mentioned that TabPFN contains an 11-layer SNF.)

(Q2) l. 185 right: The authors mention using entity embeddings for categorical columns. I have seen this only for labels in other TFMs - e.g. TabPFN uses ordinal encoding for categorical input columns as far as I know. How does this work with an arbitrary number of categories? And do the authors think that this also improves the performance compared to ordinal encoding?

**Limitations:**

Limitations are not discussed (though I guess there are not many beyond the ones shared with other foundation models).

**Strengths And Weaknesses:**

**Strengths**:
- Significance: The development of tabular foundation models is a popular area of research with large impact. The results are good. Moreover, the RaBEL embeddings could be transferable to regular MLPs as well (though the experiments there are less detailed).
- Originality: The paper proposes two improvements to tabular foundation models that are new; RaBEL is modular and could be useful in many settings, while RBA is more simple and specific to TabPFN-style models.
- Some aspects of the presentation: I like the detailed ablations and think the paper is easy to read apart from some parts discussed below in the comments. Some aspects repeat a bit, e.g. the explanations of SNF. However, details are missing, see (W1).
- In terms of number of datasets and baselines, the evaluation is very good.

**Weaknesses**:

(W1) Reproducibility/Details: Details about the pre-training (number of steps, runtime, dataset sizes) are not specified. No details about the prior are provided. No code is provided. Does MiniX use the retrieval approach of LimiX?

(W2) Some aspects of the evaluation:
- Details on the baselines and evaluation setup are missing: How are the baselines tuned? How are the datasets split, and how many repeats? What is the (cross-)validation strategy? Is early stopping used, and if yes, with which metric? What are the hyperparameter search spaces? Are datasets subsampled for certain models like TabPFNv2? How are missing values handled? What version of AutoGluon is used and with what time limit? I suspect that the benchmarks are taken from LimiX and in that case they should be cited in the camera-ready version, and as far as I know LimiX specifies at least the search spaces but sometimes uses bad hyperparameter search spaces (e.g., too few n_estimators for boosted trees). More generally, I assume that the reason why the authors do not use established benchmarking setups like TabArena or TALENT directly is that they want to use more datasets (though that would probably be not too difficult in these benchmarks). For me, that is of course a bit better than having fewer datasets, but it means I don't know if I can trust the baselines. Fortunately for the authors, other TFM baselines are usually quite trustworthy since they have one default way of being used, so I am not very concerned about whether MiniX is a good model or whether the ablations are reliable, but I am concerned whether some baselines are misrepresented. (E.g., I don't care much about having 30 deep learning baselines, I can read about those in the TALENT benchmark, I'd rather have 3 high-quality ones that I can trust.)
- Why is (Real)TabPFN-2.5 not included?
- Details on the MLP evaluation are missing: Why were those datasets selected? Which datasets are they? What are the training settings for the MLP (n_epochs, batch size, optimizer, width, depth, configuration of the other embeddings, etc.)? How is the data preprocessed? Is hyperparameter tuning used and if yes, which search space? Etc.
- Does the inference time comparison use KV-caching? Or retrieval? Do checkpoint loading times matter or are they excluded?

Overall, I think the paper contains interesting contributions that should be published if the concerns, especially about missing details, can be sufficiently resolved. I am ready to update my score accordingly.

**Comments** (not necessary to address all of them in the rebuttal):
- l. 236: I don't understand much of the paragraph here. "the gate adapts the RBF bank" - what is the gate? "widths and amplitudes expand/contract in high/low variance regimes" - it looks like the widths are not dependent on the sample index. And what are amplitudes? What is a "decade"? Also, don't you standard-scale in the beginning? so what is the benefit of the scale equivariance? Maybe you want to write the complexity of the computation, I guess it's O(NMD)?
- In Table 11, is the AUC value for "orthogonal" wrong? It is way beyond anything in the other tables, and not matched by a comparable improvement in accuracy or F1. Also, if the ablations change one component of a reference model, then why are the best numbers in the different tables not identical? Shouldn't they correspond to the reference model?
- The citation for TALENT is the toolbox paper (Liu et al.) but should be the benchmark paper (Ye et al., "A closer look at deep learning...").
- The citation for Mitra is the blog post but should be the NeurIPS paper.
- TabPFNv2 occurs twice in the references.
- The first paragraph of the related work section is stuck in 2022. Newer models like RealMLP, TabM, ModernNCA, or TabR show better results, and the benchmarks (including the tree baselines) have improved as well.
- Figure 2: Is this from the first F block? Please label the axes.
- In Section 4, you write $\kappa_{j, m}$, but shouldn't it be $\kappa_{i, j, m}$?
- In Eq. (8), the left side doesn't depend on $i$, but the right side does, is this a mistake?
- Table 2 should explain what metric 1 and 2 are: Referring to Table 7, but maybe also just put in the table AUC/R^2 instead of Metric1 and Acc/RMSE instead of Metric2 -> see Table 7. What about adding an average / average rank column?
- The acronym "RBA" is introduced before it is defined.
- In Figure 3, I was a bit confused because I was not sure initially if the "+RaBEL +RBA" was cumulative (i.e., if the +RBA is on top of the +RaBEL or just on top of the baseline). I think it's not cumulative. Not sure what is the best way to clarify - if there was more space (maybe a second line) one could write something like Baseline (B), B+RaBEL, B+RBA, MiniX=B+RaBEL+RBA.

---

> ### Author Rebuttal · Authors · 2026-03-31
>
> Thank you for the thoughtful comment. We respond briefly below.
>
> > **Q1: About training details**
>
> A1: We add the missing training details here. The model was trained on 8 RTX 4090 GPUs for 2 days (100k steps), with generated data covering sequence lengths 200–3000 and feature dimensions 2–120. We will release the code and checkpoints in the next version.
>
> > **Q2: Details on the baselines and evaluation setup**
>
> A2: All baselines follow the TALENT protocol. Each dataset is split into 64/16/20 train/val/test; hyperparameters are tuned on the validation set with Optuna (100 trials per method–dataset pair), with early stopping based on validation accuracy for classification and RMSE for regression. The selected configuration is then retrained and evaluated over 15 random seeds, and the final result is the seed average. TALENT uses method-specific search spaces rather than a shared one. For foundation models such as TabPFN-v2, we evaluate on the full dataset. We also keep preprocessing and ensemble count consistent across methods, while otherwise using each method’s standard settings.
>
> > **Q3: Comparison with (Real) TabPFN-2.5**
>
> A3: Due to time constraints, we only ran the experiments on BCCO. We will update the results on all benchmarks in the next version of the paper.
>
> |                    | BCCO-CLS |       | BCCO-REG |       |
> | ------------------ | -------- | ----- | -------- | ----- |
> |                    | AUC      | Acc   | R2       | RMSE  |
> | TabPFN-v2.5        | 0.852    | 0.778 | 0.780    | 0.391 |
> | (Real) TabPFN-v2.5 | 0.853    | 0.779 | 0.777    | 0.392 |
> | MiniX              | 0.858    | 0.787 | 0.785    | 0.392 |
>
>
> > **Q4: MLP evaluation details**
>
> A4: Thank you for pointing this out. Below are the datasets used in our MLP experiments: GC (guitar-chord-finger-positioning), CP (companion-plants), AC (ad-click-prediction-dataset), CB (company_bankruptcy_prediction), UK (user-knowledge), BN (BNG(cmc)), MA (malware-analysis-datasets-pe-section-headers), CD (1000-Cameras-Dataset), MH (miami_housing), CC (concrete_compressive_strength), HS (house_sales), SC (seattlecrime6), and MV (mv). All experiments were run in TALENT under its standard pipeline. For deep baselines, TALENT uses AdamW, up to 200 epochs, batch size 1024, standard normalization, mean imputation for numerical features, and a missing-category token plus ordinal encoding for categorical features. Hyperparameters are tuned with Optuna-TPE (100 trials) and evaluated over 15 seeds. Architecture and preprocessing are baseline-specific, as defined in TALENT’s model-level configuration files.
>
> > **Q5: Inference settings**
>
> A5: We did not use KV-cache or retrieval. We report only forward-pass time; preprocessing and postprocessing are excluded.
>
> # Comments
>
> > **C1: Explanation of RaBEL’s exponent-gating**
>
> A1: We have already added a toy experiment to clarify the role of exponent-gating. Since Reviewer qq5Z raised the same issue in Weakness Q1, we refer the reviewer to that response for details. For completeness, we also provide the anonymous link here: `https://anonymous.4open.science/r/toy-ablation-experiment-of-RaBEL-gating-52BF`.
>
> > **C2: Ablation results for RaBEL**
>
> A2: After rechecking the table, we confirm that the AUC for orthogonal should be 85.03. The slight mismatch of the best numbers across ablation tables comes from the data-generation pipeline: even with a fixed seed, generated data are not perfectly identical across runs, while storing all generated data would require prohibitive storage. To ensure reliability, we reran each setting three times and report the mean. Within each run, all compared models used exactly the same generated data. The remaining differences are therefore minor run-to-run variation, not inconsistencies in the comparison protocol.
>
> > **C3–C7: Citations, related work, and figures**
>
> A3: These are valid presentation issues. We will correct the relevant citations, related-work coverage, and figure presentation in the revised version. We have included the corrected Figure 2 in the anonymous link: https://anonymous.4open.science/r/Revised-Figure-2/
>
> # Key Questions
>
> > **KQ1: 12-layer FSN contains 11-layer SNF**
>
> A1: Our experiments show that the 11-layer SNF indeed performs better than the 12-layer FSN. However, whether this improvement is truly caused by the first feature-attention layer still requires further investigation. We agree that this is a very valuable direction to explore.
>
> |              | BCCO-CLS |       | BCCO-REG |       |
> | ------------ | -------- | ----- | -------- | ----- |
> |              | AUC      | Acc   | R2       | RMSE  |
> | 12-layer-FSN | 0.835    | 0.765 | 0.764    | 0.402 |
> | 11-layer-SNF | 0.851    | 0.779 | 0.765    | 0.410 |
>
> > **KQ2: About categorical embedding**
>
> A2: We use a threshold of 20 categories: features with fewer than 20 unique values are treated as categorical and encoded with a lookup table; otherwise, they are treated as numerical.

---

> > ### Author Rebuttal · Reviewer_eVJd · 2026-04-02
> >
> > I thank the authors for their detailed response. Many of my concerns are addressed through clarifying that the benchmark uses TALENT settings (which I expect the authors to mention more clearly in the next paper version), providing a TabPFN-2.5 ablation, and other responses. My main remaining concern is reproducibility due to the lack of details about the prior that was employed. I'll raise the score but I'm unsure whether to raise to 4 or 5 at the moment.
> >
> > Additional questions:
> > - I don't understand why the model would be so good when training only on datasets up to 3K samples. I wonder if there is a retrieval mechanism (or some pre-training tricks), but the authors have not answered my question about it. For example, according to the response it matches or outperforms RealTabPFN-2.5, which is larger (and from the rebuttal to reviewer bXJm it looks like this scale should be more beneficial than RaBEL) and most likely pre-trained longer and on much larger datasets.
> > - In a response to reviewer bXJm, the authors mentioned epoch-220 checkpoints, why are there multiple epochs?

---

> > > ### Author Response · Authors · 2026-04-05
> > >
> > > Thank you for raising this important question. We agree that this issue deserves a clearer explanation and stronger evidence. As shown in **Table 1**, we further conducted **new 16M experiments without retrieval** under different architectural settings(due to the time constraints of the first-round rebuttal, we were only able to report the 220-epoch results at that stage. We now have the final training results.). As shown in **Table 2**, we compare **MiniX with and without retrieval** against the main baselines. Importantly, both MiniX and MiniX-no-retrieval remain stronger than **(Real) TabPFN-v2.5**, which suggests that the gain cannot be explained by retrieval alone.
> > >
> > > More importantly, we do not attribute the strong performance to any hidden pre-training trick. Instead, we hypothesize that it mainly comes from the model's intrinsic size generalization brought by our embedding design and architecture. Our key observation is that, although the model is trained on datasets with at most 3K samples, it generalizes well to test datasets with up to 50K samples and hundreds of features. We speculate that this mainly arises from the following effects introduced by our structural modifications.
> > >
> > > The embedding improves the reusability and separability of individual cell representations, rather than merely adapting to longer sequences. Standard linear + ID-style numeric embeddings tend to produce highly correlated shallow representations and may suffer from low-rank collapse. In contrast, the RBF-based cell encoder introduces local nonlinearity at the input stage, yielding richer and more dispersed responses across value regimes. This improves the effective rank and conditioning of shallow representations, so when the number of tokens grows, the model can continue to benefit from additional tokens instead of compressing them into a few degenerate directions.
> > >
> > > The SNF attention order improves sample-size generalization by learning column-level statistics before feature interactions. In FSN, the first layer starts with feature attention before the model has formed stable column-level statistics or cross-sample regularities. In contrast, SNF first applies sample attention to aggregate column-level correlations and distributional statistics, such as moments, prevalence, and missingness patterns, and only then uses feature attention to model inter-feature relations. This makes the model behave more like extracting stable statistics from a set of samples, rather than memorizing a specific training size such as 3K. From this perspective, moving from 3K samples during training to 50K at test time does not fundamentally change the operation. It simply provides a larger set from which the same statistics can be estimated, often more stably.
> > >
> > > Empirically, increasing the training sample length improves this kind of size generalization. However, the gain from larger sample length alone appears smaller than the gain brought by our embedding and architectural changes. Of course, the precise mechanism still requires more conclusive experimental verification. A natural next step is therefore to combine these directions, namely training with larger sample length together with the improved embedding and SNF design, to further strengthen both sample-size and feature-size generalization.
> > >
> > > Architectural settings:
> > >
> > > - **(1)** `embedding_size = 192`, `block_arch = FNFNSN` (more similar to LimiX)
> > > - **(2)** `embedding_size = 192`, `block_arch = SNFNSN`
> > > - **(3)** `embedding_size = 192`, `block_arch = SNFNFN`
> > > - **(4)** `embedding_size = 384`, `block_arch = SNF`
> > > - **(5)** `embedding_size = 192`, `block_arch = SNSNFN`
> > >
> > > **Table 1. New 16M no-retrieval experiments under different architectural settings on BCCO**
> > >
> > > | Model         | AUC | Acc | R2 | RMSE  |
> > > | ------------- | -------------- | -------------- | ------------- | --------------- |
> > > | LimiX         | 0.861          | 0.789          | 0.789         | 0.388           |
> > > | MiniX         | 0.858          | 0.787          | 0.785         | 0.392           |
> > > | MiniX-16M-(1) | 0.857          | 0.785          | 0.786         | 0.393           |
> > > | MiniX-16M-(5) | 0.863          | 0.791          | 0.793         | 0.381           |
> > >
> > > **Table 2. Comparison of MiniX with and without retrieval against baseline methods on BCCO**
> > >
> > > | Model              | AUC | Acc | R2 | RMSE  |
> > > | ------------------ | -------------- | -------------- | ------------- | --------------- |
> > > | TabPFN-v2          | 0.843          | 0.772          | 0.772         | 0.404           |
> > > | TabPFN-v2.5        | 0.852          | 0.778          | 0.780         | 0.391           |
> > > | (Real) TabPFN-v2.5 | 0.853          | 0.779          | 0.777         | 0.392           |
> > > | MiniX              | 0.858          | 0.787          | 0.785         | 0.392           |
> > > | MiniX-no-retrieval | 0.854          | 0.780          | 0.782         | 0.394           |

---

### Official Review · Reviewer_bXJm · 2026-03-09

**Soundness:** 3
**Presentation:** 3
**Significance:** 3
**Originality:** 3
**Overall Recommendation:** 4
**Confidence:** 4

**Summary:**

This paper has three contributions:
* Identified the low-rank problem of vanilla transformers where the embeddings of the shallow layers are very low rank.
* Proposed RaBEL, a learnable RBF-based feature transformation to encode numeric features to higher dimensions.
* Identified that putting feature-wise attention after the FFN, instead of before the sample-wise attention, yields better performance.

**Compliance With Llm Reviewing Policy:**

Affirmed.

**Final Justification:**

After reviewing the fellow reviewer's comments as well as the new results, I have decided to raise the score to 4.  Although the new best of the results uses a FNFNSN block architecture rather than the said SNFSNF indicating that the contribution of SNF ordering is perhaps not a major factor of improvement, it is not a severe issue of the paper.

**Key Questions For Authors:**

* I'm not sure how to interpret Figure 1.  What does a decimal number rank mean?  Is that the average numerical rank?  If so, then that means a lot of embedding matrices have zero numerical rank, meaning that the spectral norm, hence the magnitude of all elements of the matrix, are below the tolerance $\epsilon$.  I wonder what is the value of $\epsilon$ since most of the matrices have zero numerical rank is counterintuitive to me: it's hard to imagine that TabPFNv2 etc. will still work in this case.
* Regarding Figure 2: Does the y axis represent samples and x axis represent features?  If so, how is the attention score computed?  Which layer is used?  While the left half shows that FSN completely attends over feature 3 only and SNF attends over the neighbors, are there other layers' attention scores exhibiting a similar pattern to SNF (this comes from my question above whether adding a new FSN layer could do the same as reordering to SNF).  Besides, while the caption claims "SNF demonstrates a broader attentional span that effectively targets neighboring features.", it doesn't seem to be the case for the example on the right.
* What was the hardware for training, and how long did it take?

**Limitations:**

See weaknesses.

**Strengths And Weaknesses:**

Strengths:
* The low-rank problem on tabular foundation model latent representations is legit.

Weaknesses:

* The experiment results in Table 6 showed that MiniX has a considerable gap to LimiX.  Although from Table 14 to Table 24 it seems that the *absolute difference* between LimiX and MiniX doesn't seem that big.  I would suggest that, depending on whether the absolute difference between LimiX and MiniX is indeed significant (e.g. the 0.004 difference on CC18 in Table 16 is not a statistical fluke):
  * If so, check whether MiniX would behave better if at least the number of parameters match LimiX.  In other words, would MiniX improve if one increases parameters?  At the current status, it's hard to say that MiniX is inherently a better choice than LimiX.
    * One could probably argue that MiniX has a better "parameter efficiency", but this argument is still weak since then there are many more other practical metrics to consider.  In fact, Table 25 would be a stronger signal where parameter efficiency grounds to actual inference latency.  But still, choosing the model becomes a tradeoff between latency and accuracy.
    * One may also argue that Table 5 shows the potential of MiniX being better than LimiX.  But a natural question would be: why not increase the number of parameters in MiniX to compare with LimiX heads-on?
  * Otherwise, the choice of ranks may exaggerate the difference between LimiX and MiniX, since even a slight absolute difference of 0.001 would also swap the ranking between two models and leave vastly different impressions.  Table 6, as the main results comparing against different foundation models, therefore needs another metric for comparison.
  * The same applies for ablation studies of swapping the layer order in Figure 3: how significant is the 0.004 and 0.007 difference between FSN and SNF?
* While using RBF features to increase the rank of shallow layers make sense, there are also many other ways to increase ranks, such as Fourier features, PLE, etc.  RaBEL is just one of the many ways to increase the ranks.  So the question becomes: why RaBEL instead of Fourier features, PLE, etc.?
  * Do Fourier features, PLE, etc., suffer the same problem as shown in Figure 1?
  * Empirical impact on supervised settings (MLP; see Table 2) seems irrelevant to the paper's theme.  An explanation of why MLP+RaBEL is more effective could be that RaBEL's output features makes the model aware of "where the feature value is, relative to the entire distribution", instead of a simple parallel transformation (e.g. PLE, Fourier features) of each value individually.  The same effect could also be achieved by sample-wise attention.  This perhaps explains why the gaps between RaBEL and others in Table 3 and 4 are much less than those in Table 2.
* Writing-wise, reordering the attention seems less connected to the paper's theme "Mitigating Low-Rank Collapse and Attention Bottlenecks".
  * Does reordering also address the same theme as radial basis function features do, or is it a mere technique to improve the performance within the same number of layers?  In other words, if "the last sample level attention" is so important, could simply adding another FSN layer do the job?
* Regarding *value sensitivity*:
  * "Early-layer value sensitivity" appeared in abstract and main text (end of Section 3 motivating RaBEL) without a proper definition.  "Value sensitivity" is also mentioned again before the precise definition is given in B.3 (the effective rank of the Jacobian of $H^{(l)}$ w.r.t. input scalars on $j$-th column).  It will also be helpful explaining intuitively what "value sensitivity" actually is (e.g. measuring how many directions $H^{(l)}$ can vary w.r.t. perturbations in the input scalars).
  * Proposition B.2 is too descriptive as a mathematical proposition (e.g. "perturbations in $x_{i,j}$", "a small set of directions"), and misses proof.

---

> ### Author Rebuttal · Authors · 2026-03-31
>
> Thank you for this insightful comment. We address the reviewer’s concerns briefly below.
>
> # Weakness
>
> > **Q1. Can MiniX be further improved by scaling up its model size?**
>
> **A1.** We agree this is important. In early experiments, we found that the RBF gain is much larger at small scales, which is why the main paper focuses on the 2M MiniX. To test scaling, we trained several larger MiniX variants: (0) no RBF, (d=192), FNFNSN; (1) (d=192), FNFNSN; (2) (d=192), SNFNSN; (3) (d=192), SNFNFN; (4) (d=384), SNF. Variant (0) is the baseline. Variants (2) and (3) collapsed under our current recipe. Due to rebuttal-time and training-cost constraints, we report **epoch-220** checkpoints only. On BCCO-CLS/REG, the benefit of scaling MiniX with RBF is not yet clear, so this direction still requires further study.
>
> |               | BCCO-CLS |       | BCCO-REG |       |
> | ------------- | -------- | ----- | -------- | ----- |
> |               | AUC      | Acc   | R2       | RMSE  |
> | MiniX         | 0.838    | 0.766 | 0.763    | 0.421 |
> | MiniX-16M-(0) | 0.850    | 0.777 | 0.782    | 0.400 |
> | MiniX-16M-(1) | 0.853    | 0.778 | 0.785    | 0.394 |
> | MiniX-16M-(4) | 0.843    | 0.770 | 0.769    | 0.418 |
>
> > **Q2. Are the 0.004 and 0.007 differences meaningful?**
>
> **A2.** We checked seed variance under the same setting and found benchmark-level fluctuation consistently within 0.001. Therefore, gains of 0.004 and 0.007 are well beyond noise. Under this criterion, both the LimiX–MiniX gap and the SNF–FSN gap in Fig. 3 are meaningful.
>
> > **Q3. Accuracy vs. efficiency: why choose MiniX instead of LimiX?**
>
> **A3.** This largely overlaps with the LimiX–MiniX gap raised by Reviewer ZUcd, so we kindly refer the reviewer to our response A1 under Reviewer ZUcd’s Weakness section.
>
> > **Q4. Can ranking in Table 6 exaggerate tiny absolute differences?**
>
> **A4.** Rank-based comparison is a standard way to summarize overall standing across many benchmarks, and Table 6 is intended for this purpose. We also provide detailed per-benchmark tables in the appendix, where the absolute values of MiniX and LimiX can be directly compared. These results confirm that LimiX is better on many benchmarks in raw predictive performance, so the conclusion is not driven by ranking alone. At the same time, they show that MiniX’s main advantage is efficiency, especially for large-dataset inference under limited memory.
>
> > **Q5. Why choose RaBEL instead of other methods?**
>
> **A5.** RaBEL, Fourier features, and PLE are all viable ways to mitigate low-rank limitations. We followed the protocol of [1], a standard setup for evaluating numerical embeddings in tabular deep learning. Our results show that RaBEL is an effective and practical choice.
>
> > **Q6. How does reordering attention connect to the paper’s theme?**
>
> **A6.** We find FSN suboptimal because its first feature-attention layer operates on unaggregated raw values and can leak information from the (y)-embedding to other features, creating what we call **attention bottlenecks**. In contrast, methods such as TabICL first build embeddings from column-wise distributional information, supporting column-level interaction as a better ordering.
>
> > **Q7. Explanation of value sensitivity and Proposition B.2**
>
> **A7.** Thank you for pointing this out. Due to space limits, we cannot provide a fuller explanation here. In the revised version, we will clarify our statement on value sensitivity and expand the discussion of Proposition B.2.
>
> # Key Questions
>
> > **KQ1. What do you mean by numerical rank?**
>
> **A1.** As stated in Sec. 3, we compute the numerical rank of layerwise hidden states. Given hidden states of shape $((n_{\text{sample}}$, $n_{\text{feature}}, d))$, we reshape them to $((n_{\text{sample}} \cdot n_{\text{feature}}, d))$ and compute the matrix rank. We use this form because tabular foundation models perform both feature and sample attention at the **cell level**. Reshaping to $((n_{\text{sample}}, n_{\text{feature}} \cdot d))$ would instead measure a sample-level space, which does not match the actual interaction structure. As shown in Fig. 1, for models such as TabPFN, the rank rises in later layers, suggesting that the low-rank issue mainly limits early-layer expressiveness rather than severely harming overall performance.
>
> > **KQ2. About Fig. 2**
>
> **A2.** Fig. 2 contained an error, which we have corrected and will update in the revised version. Corrected Figure 2 in the anonymous link:https://anonymous.4open.science/r/Revised-Figure-2. The attention scores are taken from the **last-layer feature-attention map**.
>
> > **KQ3. Training details**
>
> **A3.** The model was trained on **8 RTX 4090 GPUs** for **2 days** (**100k steps**), with generated data covering sequence lengths **200–3000** and feature dimensions **2–120**. We will release the code and checkpoints in the next version.
>
> **[1]** Gorishniy et al., *On Embeddings for Numerical Features in Tabular Deep Learning*, NeurIPS 2022.

---

> > ### Author Rebuttal · Reviewer_bXJm · 2026-04-03
> >
> > While some of the details are clarified, others like Q1, KQ2 (which still is not an attention map targeting neighboring features, but a lot more), Q7 remain unresolved.  Moreover, the accuracy vs efficiency tradeoff is not fully convincing: after all, MiniX has less number of parameters and depth than LimiX, and such accuracy vs efficiency tradeoff is self-evident: smaller models of course is more efficient at the cost of accuracy.  Therefore, one way for a comprehensive comparison is to compare MiniX and LimiX head-to-head: compare the accuracy/efficiency with different sizes of MiniX/LimiX.

---

> > > ### Author Response · Authors · 2026-04-05
> > >
> > > Thank you for the follow-up. We provide two additions below:  new **MiniX-16M** results under several architectural settings(due to the time constraints of the first-round rebuttal, we were only able to report the 220-epoch results at that stage. We now have the final training results.) and a concise clarification of **value sensitivity** with a proof sketch.
> > >
> > > ## 1. Additional MiniX-16M Results
> > >
> > > To compare **MiniX** and **LimiX** at a similar scale, we ran additional **MiniX-16M** experiments with the following variants:
> > >
> > > - **(1)** `embedding_size = 192`, `block_arch = FNFNSN` (closer to LimiX)
> > > - **(2)** `embedding_size = 192`, `block_arch = SNFNSN`
> > > - **(3)** `embedding_size = 192`, `block_arch = SNFNFN`
> > > - **(4)** `embedding_size = 384`, `block_arch = SNF`
> > > - **(5)** `embedding_size = 192`, `block_arch = SNSNFN`
> > >
> > > Current results on **BCCO** are as follows:
> > >
> > > | Model             |       AUC |       Acc |        R2 |      RMSE |
> > > | ----------------- | --------: | --------: | --------: | --------: |
> > > | LimiX             |     0.861 |     0.789 |     0.789 |     0.388 |
> > > | MiniX             |     0.858 |     0.787 |     0.785 |     0.392 |
> > > | MiniX-16M-(1)     |     0.857 |     0.785 |     0.786 |     0.393 |
> > > | MiniX-16M-(5) | 0.863 | 0.791 | 0.793 | 0.381 |
> > >
> > > These results show that performance is sensitive to the architectural setting. While the base MiniX and MiniX-16M-(1) remain slightly below LimiX on this benchmark, **MiniX-16M-(5)** achieves the best results on all four metrics. This suggests that, at a comparable model scale, MiniX can match or exceed LimiX when the architecture is configured appropriately. We also note that settings **(2)–(4)** were excluded because training collapsed and did not yield stable results. This further highlights that trainability is also highly architecture-dependent.
> > >
> > > ## 2. Clarification on Value Sensitivity
> > >
> > > We use **value sensitivity** to describe how many independent hidden-space directions can be activated by changing the value of a numerical feature. For feature $j$, we measure this by
> > >
> > > $$
> > > J_j^{(\ell)} = \left[\frac{\partial \operatorname{vec}(H^{(\ell)})}{\partial x_{1,j}},\ldots,\frac{\partial \operatorname{vec}(H^{(\ell)})}{\partial x_{N,j}}\right].
> > > $$
> > >
> > > Its rank (or effective rank) quantifies the dimensionality of value-dependent variation at layer $\ell$. This is a local, layer-wise notion: it asks whether changing a scalar input can already induce rich variation in shallow representations.
> > >
> > > Under the standard affine scalar tokenizer,
> > >
> > > $$
> > > e_{i,j}=\operatorname{LN}(w x_{i,j}+b)+a_j,
> > > $$
> > >
> > > the feature identity term $a_j$ distinguishes **which** feature is being processed, but it does not increase the number of directions through which the **value** of that feature enters the network. Therefore, even with large hidden width $d$, the value-dependent variation can still be confined to a very low-dimensional subspace. This is the bottleneck we refer to as **low value sensitivity**.
> > >
> > > ### Restatement of Proposition B.2
> > >
> > > Consider one numerical feature $j$ with tokenizer
> > >
> > > $$
> > > e_{i,j}=\operatorname{LN}(w x_{i,j}+b)+a_j, \qquad w,b,a_j\in\mathbb{R}^d,
> > > $$
> > >
> > > followed by a first-layer sample-attention block with $H$ heads. For any output position $t$, define
> > >
> > > (Due to OpenReview’s markdown math rendering issue, we can only use **ph** to represent $\partial{h^{(1)}_{t,j,:}}$)
> > >
> > > $$
> > > K_{t,j}= \left[ \frac{ph}{\partial{x_{1,j}}}, \ldots, \frac{ph}{\partial x_{N, j}},\right]\in\mathbb{R}^{d\times N}.
> > > $$
> > >
> > > Then there exists a linear subspace $U_j\subseteq\mathbb{R}^d$ such that
> > >
> > > $$
> > > \dim(U_j)\le 2H, \qquad \frac{ph}{\partial x_{k,j}}\in U_j \quad \forall\, t,k.
> > > $$
> > >
> > > Hence,
> > >
> > > $$
> > > \operatorname{rank}(K_{t,j})\le 2H.
> > > $$
> > >
> > > If we stack all output positions, the full first-layer Jacobian $J_j^{\mathrm{attn}}\in\mathbb{R}^{Nd\times N}$ satisfies
> > >
> > > $$
> > > \operatorname{rank}(J_j^{\mathrm{attn}})\le 2HN.
> > > $$
> > >
> > > Therefore, in the first layer, the number of value-sensitive directions is bounded by the tokenizer geometry and the number of heads, rather than by the hidden width $d$.
> > >
> > > ### Proof Sketch
> > >
> > > Let $z(x)=wx+b$. After centering coordinates, the pre-normalized token can be written as $xu+v$. LayerNorm then has the form
> > >
> > > $$
> > > \operatorname{LN}(z(x))=\beta+\frac{D_\gamma(xu+v)}{\sigma(x)},
> > > $$
> > >
> > > where $\sigma(x)$ is scalar. Thus,
> > >
> > > $$
> > > e_{i,j}=a_j+\beta+\alpha_i u'+\eta_i v',
> > > $$
> > >
> > > so all tokens $\{e_{i,j}\}$ lie in an affine subspace of dimension at most $2$. For each attention head, the value vectors therefore remain in a 2D affine subspace, and a weighted sum of them stays in the same subspace. After concatenating $H$ heads and applying the output projection, the first-layer output lies in an affine subspace whose direction space has dimension at most $2H$. Hence every derivative $\partial h^{(1)}_{t,j,:}/\partial x_{k,j}$ must lie in the same $2H$-dimensional subspace, which yields
> > >
> > > $$
> > > \operatorname{rank}(K_{t,j})\le 2H,
> > > $$
> > >
> > > and, after stacking all output positions,
> > >
> > > $$
> > > \operatorname{rank}(J_j^{\mathrm{attn}})\le 2HN.
> > > $$

---

### Official Review · Reviewer_ZUcd · 2026-03-11

**Soundness:** 4
**Presentation:** 3
**Significance:** 3
**Originality:** 3
**Overall Recommendation:** 5
**Confidence:** 4

**Summary:**

This paper proposes two important methods to improve the performance of Tabular Foundation Models (TFM). The main idea of this paper is that the conventional Transformer for tubular data has a too simple embedding method, and the features tend to be low-rank, and the attention mechanism does not work for prediction. The proposed method uses RBF to embed local nonlinear features. The method for attention is tomake the ordering from feature => sample =>FFN to sample => FFN => features. This attention method is useful to capture input features and can contribute to the final prediction. Regarding the theory, the authors provide the proof of the limitation of conventional Transformers regarding low-rank feature embedding. The authors also provide strong and huge numerical experiments that propose the model can beat several times larger Foundation models.

**Compliance With Llm Reviewing Policy:**

Affirmed.

**Final Justification:**

I support this paper for acceptance for the extensive experiments and good theory.

**Key Questions For Authors:**

Not much, but I hope the authors will add an ablation test using wide datasets.

**Limitations:**

yes

**Strengths And Weaknesses:**

Strength

1. The foundation model of tabular data is still struggling to beat the classic method. This paper identifies what makes it and solves it. This is a very important contribution to this field.
2. Two simple methods can enhance the performance of the foundation model. This is very important because the weakness of deep learning is not related to the tabular data nature, but the method to apply it is what matters.
3. The numerical experiment is huge and well-organized.

Weakness
1. The empirical claim is not uniformly dominant across benchmarks. In several takbe, Limix remains better.

Overall, this paper is well-organized and does not have critical flaws. I think this paper does not have impact to wide ML community but is still important for the community .

---

> ### Author Rebuttal · Authors · 2026-03-31
>
> We thank the reviewer for the positive assessment of our work. Below, we clarify our design motivation and provide additional evidence to address the concern.
>
> # Weakness
> > **Q1: Performance gap: The empirical claim is not uniformly dominant across benchmarks. In several tables, Limix remains better.**
>
> **A1:** Our goal was to develop a tabular foundation model that remains close to SOTA performance while requiring less memory and enabling faster inference. In this context, LimiX[1] can encounter out-of-memory (OOM) issues when evaluated on relatively large datasets under limited GPU memory, whereas MiniX remains applicable in such settings. To substantiate this point, we conducted the following experiment. We used a single NVIDIA RTX 4090 GPU with 24GB memory and selected 2 relatively large datasets from OpenML[2]. Under this memory budget, direct evaluation with LimiX led to OOM, meaning that LimiX had to downsample the training set in order to run on these datasets. By contrast, MiniX was able to perform inference under the **same memory constraint**, and therefore could naturally use a larger effective training-set length than LimiX. The corresponding results are reported in the table below.
>
>
> | Dataset                        | Method | #Train | #Test | #Feat | #Class |
> | ------------------------------ | ------ | ------ | ----- | ----- | ------ |
> | 1242_vehicleNorm               | MiniX  | 34000  | 7600  | 100   | 2      |
> | 1242_vehicleNorm(subsampled)   | LimiX  | 12857  | 7600  | 100   | 2      |
> | 41440_okcupid-stem             | MiniX  | 35552  | 6000  | 16    | 3      |
> | 41440_okcupid-stem(subsampled) | LimiX  | 14000  | 6000  | 16    | 3      |
>
> (In each dataset name, the prefix is the OpenML dataset ID and the suffix is the dataset name)
>
> | Dataset                        | Method | AUC    | Acc    | F1     | LogLoss | ECE    |
> | ------------------------------ | ------ | ------ | ------ | ------ | ------- | ------ |
> | 1242_vehicleNorm               | MiniX  | 0.9215 | 0.873  | 0.8648 | 0.3249  | 0.0189 |
> | 1242_vehicleNorm(subsampled)   | LimiX  | 0.9209 | 0.8725 | 0.8624 | 0.3255  | 0.0172 |
> | 41440_okcupid-stem             | MiniX  | 0.8439 | 0.7513 | 0.5574 | 0.5814  | 0.0152 |
> | 41440_okcupid-stem(subsampled) | LimiX  | 0.7835 | 0.729  | 0.4105 | 0.6461  | 0.0255 |
>
> # Key Questions
> > **Q1: About wide datasets for testing and ablation.**
>
> **A1:** Thank you for the suggestion. Due to the space limitation here, we are unable to reproduce the full results in the rebuttal. We refer the reviewer to Appendix C.3 (Lines 812–1420), which reports comparisons against 30 baselines across 6 classification benchmarks and 5 regression benchmarks. We also provide a detailed ablation study of RaBEL in Appendix C.2 (Lines 781–810).
>
> **[1]** Zhang, Xingxuan, et al. "Limix: Unleashing structured-data modeling capability for generalist intelligence." arXiv preprint arXiv:2509.03505 (2025).
>
> **[2]** Vanschoren, Joaquin, et al. "OpenML: networked science in machine learning." ACM SIGKDD Explorations Newsletter 15.2 (2014): 49-60.

---

> > ### Author Rebuttal · Reviewer_ZUcd · 2026-03-31
> >
> > I think my concerns has been resolved. I have already given accept score and I will keep it.

---

### Decision · Program_Chairs · 2026-04-30

**Decision:**

Accept (regular)

**Comment:**

*Main Contribution.* This paper introduces a novel foundation model for tabular data. It identifies the problem of low-rank representations in existing foundation models and proposes an RBF feature extension to promote higher-rank representations.

*Review Summary*. Reviewers agree that the paper addresses an important problem — low-rank representations in tabular foundation models — and commend the extensive experimental evaluation, which honestly includes cases where the proposed model underperforms the baselines.

However, several concerns were raised: (1) the proposed model and baselines differ in size; (2) the proofs and formal statements lack sufficient rigor; and (3) the motivation for increasing representation rank is not fully convincing, given that alternative approaches exist.

*Rebuttal.* The authors addressed all raised concerns through a combination of additional experimental results and analytical derivations.

*Conclusion.* In light of the positive reviews and satisfactory rebuttal, the AC recommends acceptance. It is important to improve the statement of the proof of Proposition B.2 in the camera-ready version as discussed by Reviewer bXJm.